# REMEMBER AND FORGET FOR EXPERIENCE REPLAY

## ABSTRACT

Experience replay (ER) is crucial for attaining high data-efficiency in off-policy deep reinforcement learning (RL). ER entails the recall of experiences obtained in past iterations to compute gradient estimates for the current policy. However, the accuracy of such updates may deteriorate when the policy diverges from past behaviors, possibly undermining the effectiveness of ER. Previous off-policy RL algorithms mitigated this issue by tuning their hyper-parameters in order to abate policy changes. We propose `ReF-ER`, a method for active management of experiences in the Replay Memory (RM). `ReF-ER` forgets experiences that would be too unlikely with the current policy and constrains policy changes within a trust region of the behaviors in the RM. We couple `ReF-ER` with Q-learning, deterministic policy gradient and off-policy gradient methods to show that `ReF-ER` reliably improves the performance of continuous-action off-policy RL. We complement `ReF-ER` with a novel off-policy actor-critic algorithm (`RACER`) for continuous-action control. `RACER` employs a computationally efficient closed-form approximation of the action values and is shown to be highly competitive with state-of-the-art algorithms on benchmark problems, while being robust to large hyper-parameter variations.

## 1   INTRODUCTION

Deep Reinforcement Learning (RL) has an ever increasing number of success stories ranging from realistic simulated environments (Schulman et al., 2015; Mnih et al., 2016), robotics (Levine et al., 2016) and games (Mnih et al., 2015; Silver et al., 2016). Experience Replay (ER) (Lin, 1992) enhances deep RL algorithms by using information collected in past policy ($\mu$) iterations to compute updates for the current policy ($\pi$). ER has become one of the mainstay techniques to improve the sample-efficiency of off-policy deep RL. Sampling from a Replay Memory (RM) stabilizes stochastic gradient descent (SGD) by disrupting temporal correlations and extracts information from useful experiences over multiple updates (Schaul et al., 2015b). However, when $\pi$ is parameterized by a Neural Network (NN), SGD updates may result in significant changes to the policy, thereby shifting the distribution of states observed from the environment. In this case sampling the RM for further updates may lead to incorrect gradient estimates, therefore deep RL methods must account for and limit the dissimilarity between $\pi$ and behaviors in the RM. Previous works employed trust region methods to bound policy updates (Schulman et al., 2015; Wang et al., 2017). Despite several successes, deep RL algorithms are known to suffer from instabilities and exhibit high-variance of outcomes (Islam et al., 2017; Henderson et al., 2017), especially continuous-action methods employing the stochastic (Sutton et al., 2000) or deterministic (Silver et al., 2014) Policy Gradients (PG or DPG).

In this work we redesign ER in order to control the distance between the behaviors $\mu$ used to compute the update and the current policy $\pi$. More specifically, we classify experiences either as "near-policy" or "far-policy", depending on the ratio $\rho$ of probabilities of selecting the associated action with $\pi$ and that with $\mu$. The weight $\rho$ appears in many estimators that are used with ER such as the off-policy policy gradients (off-PG) (Degris et al., 2012) and the off-policy return-based evaluation algorithm Retrace (Munos et al., 2016). From this classification, we introduce measures to limit the fraction of "far-policy" samples in the RM, as well as computing gradient estimates only from "near-policy" experiences. Furthermore, these hyper-parameters can be gradually annealed during training to obtain increasingly accurate updates from nearly on-policy experiences. Remember and Forget Experience Replay (`ReF-ER`) is a simple algorithm that can be applied to virtually any off-policy RL algorithm with parameterized policies. We show that `ReF-ER` allows better stability and performance than conventional ER in all three main classes of continuous-actions off-policy deep RL algorithms:

methods based on the DPG (ie. `DDPG` (Lillicrap et al., 2016)), methods based on Q-learning (ie. `NAF` (Gu et al., 2016)), and with off-PG (Degris et al., 2012; Wang et al., 2017).

In recent years, there has been a growing interest in coupling RL with high-fidelity physics simulations (Reddy et al., 2016; Novati et al., 2017; Colabrese et al., 2017; Verma et al., 2018). The computational cost of these simulations calls for data-efficient RL methods that are reliable and do not require problem-specific tweaks to the hyper-parameters. Moreover, while on-policy training of simple policy architectures has been shown to be sufficient in some benchmark environments (Rajeswaran et al., 2017), agents aiming to solve complex problems with highly non-linear dynamics might require deep or recurrent models that can be trained more efficiently with off-policy methods. We address these issues by introducing a simple and computationally efficient off-policy actor-critic architecture (`RACER`) for continuous-action control problems. We systematically analyze a wide range of hyper-parameters on the OpenAI Gym (Brockman et al., 2016) robotic tasks, and show that `RACER` combined with `ReF-ER` reliably obtains results that are competitive with the state-of-the-art.

## 2 PRELIMINARIES

Consider the sequential decision process of an agent aiming to optimize its interaction with the environment. At each step $t$, the agent observes its state $s_t \in \mathbb{R}^{d_S}$, performs an action by sampling a policy $a_t \sim \mu(a|s_t) \in \mathbb{R}^{d_A}$, and transitions to a new state $s_{t+1} \sim \mathcal{D}(s|a_t, s_t)$ with reward $r_{t+1} \in \mathbb{R}$. These interactions $\{s_t, a_t, r_t, \mu_t\}$ are stored in a RM, which constitutes the data used by off-policy RL to train the parametric policy $\pi^{\mathtt{w}}(a|s)$. The importance weight $\rho_t = \pi^{\mathtt{w}}(a_t|s_t)/\mu_t$ is the ratio between the probability of selecting $a_t$ with the current $\pi^{\mathtt{w}}$ and with the behavior $\mu_t$, which gradually becomes dissimilar from $\pi^{\mathtt{w}}$ as the latter is trained. The on-policy state-action value $Q^\pi(s, a)$ measures the expected returns from $(s, a)$ following the policy $\pi^{\mathtt{w}}$:

$$Q^\pi(s, a) = \mathbb{E}_{s_t \sim \mathcal{D}, \, a_t \sim \pi^{\mathtt{w}}} \left[ \sum_{t=0}^\infty \gamma^t r_{t+1} \, \middle| \, s_0 = s, a_0 = a \right] \tag{1}$$

Here $\gamma$ is a discount factor. The value of state $s$ is the on-policy expectation: $V^\pi(s) = \mathbb{E}_{a \sim \pi}[Q^\pi(s, a)]$ and the action advantage is $A^\pi(s, a) = Q^\pi(s, a) - V^\pi(s)$, such that $\mathbb{E}_{a \sim \pi}[A^\pi(s, a)] := 0$. We consider algorithms that train parametric approximators $Q^{\mathtt{w}}$ from off-policy data. The Q-learning target is $\hat{q}_t = r_{t+1} + \gamma \mathbb{E}_{a' \sim \pi} Q^{\mathtt{w}}(s_{t+1}, a')$. The Retrace algorithm (Munos et al., 2016) includes all the rewards obtained by the behavior $\mu_t$ in the value estimation:

$$\hat{Q}_t^{\text{ret}} = r_{t+1} + \gamma V^{\mathtt{w}}(s_{t+1}) + \gamma \min\{1, \rho_{t+1}\} \left[ \hat{Q}_{t+1}^{\text{ret}} - Q^{\mathtt{w}}(s_{t+1}, a_{t+1}) \right] \tag{2}$$

The off-PG (Degris et al., 2012) can be used to update $\pi^{\mathtt{w}}$ by ER: $\hat{g}_t^{\text{off-PG}}(\mathtt{w}) = \rho_t \hat{A}_t^\pi \nabla_{\mathtt{w}} \log \pi^{\mathtt{w}}(a_t|s_t)$. Here, $\hat{A}_t^\pi$ is an estimator for the on-policy advantage, such as $\hat{A}_t^{\text{ret}} := \hat{Q}_t^{\text{ret}} - V^{\mathtt{w}}(s_t)$.

## 3 REMEMBER AND FORGET EXPERIENCE REPLAY

In off-policy RL it is common to maximize on-policy returns averaged over the distribution of states in a RM (Degris et al., 2012). However, as $\pi^{\mathtt{w}}$ gradually shifts away from previous behaviors $\mu_t$, the distribution of states in the RM is increasingly dissimilar from the on-policy distribution, and trying to increase an off-policy performance metric may not improve on-policy outcomes. This issue may be compounded with algorithm-specific concerns. For example, in `ACER` (Wang et al., 2017) the dissimilarity between $\mu_t$ and $\pi^{\mathtt{w}}$ may cause vanishing or diverging importance weights $\rho_t$, thereby increasing the variance of the off-PG and deteriorating the convergence speed of Retrace by inducing "trace-cutting" (Munos et al., 2016). As a remedy, `ACER` tunes the learning rate and uses a target network (Mnih et al., 2015), updated as a delayed copy of the policy network, to constrain policy updates. Target networks are also employed in `DDPG` (Lillicrap et al., 2016) in order to slow down the feedback loop between value-network and policy-network optimizations. This feedback loop causes overestimated action values that can only be corrected by acquiring new on-policy samples. Recent works (Henderson et al., 2017) have shown the opaque variability of outcomes of continuous-action deep RL algorithms depending on hyper-parameters. Target-networks may be one of the sources of this unpredictability. In fact, when using deep approximators, there is no guarantee that the small weight changes imposed by target-networks correspond to small changes in the network's output.

Here we propose a set of three simple rules, collectively referred to as Remember and Forget ER (`ReF-ER`), to directly control the degree of "off-policyness" of the samples in the RM:

1. Updates are computed from mini-batches drawn uniformly from a RM. We compute the importance weight $\rho_t$ of each sample and classify it as "near-policy" if $1/c_{max} < \rho_t < c_{max}$ with $c_{max} > 1$. The samples with vanishing ($\rho_t < 1/c_{max}$) or exploding ($\rho_t > c_{max}$) importance weights are classified as "far-policy". When computing off-policy estimators with finite batch-sizes, such as $\hat{Q}^{ret}$ or the off-PG, "far-policy" samples may either be irrelevant or increase the variance. `ReF-ER` allows computing gradient estimates exclusively from "near-policy" samples. In order to efficiently approximate the number of far-policy samples in the RM, we store for each step its most recent $\rho_t$.

2. When acquiring samples from the environment, older episodes with the highest fraction of far-policy samples are removed until the number $n_{obs}$ of samples in the RM is at most $N$.

3. Policy updates are penalized in order to attract the current policy $\pi^w$ towards $\mu_t$:

$$\hat{g}_t^{\text{ReF-ER}}(w) = \begin{cases} \beta \, \hat{g}_t(w) - (1-\beta)\nabla D_{KL}\left[\mu_t \| \pi^w(\cdot|s_t)\right] & \text{if } 1/c_{max} < \rho_t < c_{max} \\ -(1-\beta)\nabla D_{KL}\left[\mu_t \| \pi^w(\cdot|s_t)\right] & \text{otherwise} \end{cases} \tag{3}$$

Here $D_{KL}$ is the Kullback–Leibler divergence and the coefficient $\beta \in [0,1]$ is updated after each gradient step such that a fixed fraction $D \in (0,1)$ of the RM are far-policy samples:

$$\beta \leftarrow \begin{cases} (1-\eta)\beta & \text{if } n_{far}/n_{obs} > D \\ (1-\eta)\beta + \eta, & \text{otherwise} \end{cases} \tag{4}$$

where $\eta$ is the learning rate and $n_{far}$ is the number of far-policy samples. Note that iteratively updating $\beta$ with Eq. 4 has fixed points in 0 for $n_{far}/n_{obs} > D$ and in 1 otherwise.

For $c_{max} \rightarrow 1$ and $D \rightarrow 0$, `ReF-ER` becomes asymptotically equivalent to computing updates from on-policy samples. `ReF-ER` aims to reduce the sensitivity on the network architecture and hyper-parameters by controlling the rate at which the policy can deviate from the behaviors in the RM.

## 4 RACER: REGULARIZED ACTOR-CRITIC WITH EXPERIENCE REPLAY

This work analyzes `ReF-ER` with three types of continuous-actions off-policy deep RL algorithms: methods based on the DPG (*i.e.* `DDPG`), based on Q-learning (*i.e.* `NAF`), and based on the off-PG (e.g. `ACER` (Wang et al., 2017) and `IMPALA` (Espeholt et al., 2018)). Here we introduce `RACER`, an off-PG based architecture with some improvements over `ACER` and `IMPALA` which aid our analysis of `ReF-ER`. First, `RACER` does not require expensive network architectures (easing reproducibility and exploration of the hyper-parameters). Second, it samples time steps rather than episodes (like `DDPG` and `NAF` and unlike `ACER` and `IMPALA`), further reducing its cost. Third, it does not introduce any technique that would interfere with `ReF-ER` and affect its analysis. Specifically, `ACER` uses the *variance truncation and bias correction trick* (TBC), employs a target network to bound policy

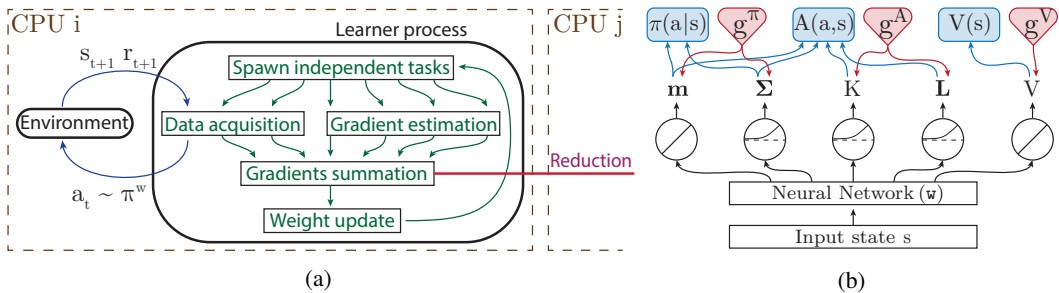

Figure 1: (a) Diagram of asynchronous ER-based RL algorithms. (b) Neural network architecture employed by `RACER`. Blue arrows connect each output with the elements of the actor-critic framework for which it is used. Red arrows represent the flow of the error signals.

updates with a trust-region scheme, and modifies Retrace to use $\sqrt[d_A]{\rho}$ instead of $\rho$. Lacking these techniques, we expect RACER to require ReF-ER to deal with unbounded importance weights.

RACER's network architecture consists of a single Multilayer Perceptron (MLP) with weights $\mathbf{w}$ which receives the state $s$ and outputs as shown in Fig. 1b. From the final layer, the first $2d_A$ outputs are taken as the mean $\mathbf{m}$ and diagonal covariance $\boldsymbol{\Sigma}$ of the Gaussian policy $\pi^{\mathbf{w}}(a|s)$. One output estimates the state value $V^{\mathbf{w}}$, and the remaining outputs are the parameters of a closed-form approximator for the action advantage $A^{\mathbf{w}}$. RACER can be extended for problems where the state is received as a visual feed (by inserting convolutional layers), or where the state is only partially observable (by substituting the MLP with a recurrent model). However, these extensions are outside the scope of this paper.

The parameters $\mathbf{w}$ are updated from mini-batches of off-policy time steps with ReF-ER. A separate gradient is defined for each component of the actor-critic framework, as sketched in figure 1b. The policy statistics $\mathbf{m}(s)$ and $\boldsymbol{\Sigma}(s)$ are updated with the off-PG (Degris et al., 2012):

$$\hat{\mathbf{g}}_t^{\pi}(\mathbf{w}) = \rho_t \hat{A}_t^{\text{ret}} \nabla_{\{\mathbf{m},\,\boldsymbol{\Sigma}\}} \log \pi^{\mathbf{w}}(a_t \mid s_t) \tag{5}$$

We estimate the on-policy advantage with Retrace. From TBC and Retrace, we obtain an estimator for the on-policy state value $\hat{V}_t^{\text{tbc}} = V^{\mathbf{w}}(s_t) + \min\{1, \rho_t\}(\hat{Q}_t^{\text{ret}} - Q^{\mathbf{w}}(s_t, a_t))$, used as a target for $V^{\mathbf{w}}$:

$$\hat{\mathbf{g}}_t^V(\mathbf{w}) = \hat{V}^{\text{tbc}}(s) - V^{\mathbf{w}}(s_t) = \min\{1,\,\rho_t\}\left[\hat{Q}_t^{\text{ret}} - Q^{\mathbf{w}}(s_t, a_t)\right] \tag{6}$$

Rather than having a separate MLP with inputs $(s, a)$ to parameterize $Q^{\mathbf{w}}$ (as in ACER or DDPG), whose expected value under the policy would be computationally demanding to compute, we employ a closed-form equation for $A^{\mathbf{w}}$ inspired by NAF (Gu et al., 2016). The network outputs the coefficients of a concave function $f^{\mathbf{w}}(s, a)$ which is chosen such that its maximum coincides with the mean of the policy $\mathbf{m}(s)$, and such that it is possible to derive analytical expectations for $a \sim \pi^{\mathbf{w}}$. In App. B we explore multiple choices of $f^{\mathbf{w}}(s, a)$. For all results in the main text of the manuscript we used:

$$f^{\mathbf{w}}(s, a) = K(s)\ \exp\left[-\frac{1}{2}\mathbf{a}_+^{\intercal}\ \mathbf{L}_+^{-1}(s)\ \mathbf{a}_+ - \frac{1}{2}\mathbf{a}_-^{\intercal}\ \mathbf{L}_-^{-1}(s)\ \mathbf{a}_-\right] \tag{7}$$

Here $\mathbf{a}_- = \min\left[a - \mathbf{m}(s), \mathbf{0}\right]$ and $\mathbf{a}_+ = \max\left[a - \mathbf{m}(s), \mathbf{0}\right]$ (both element-wise operations). This parameterization requires one MLP output for $K(s)$ and $d_A$ outputs for each diagonal matrix $\mathbf{L}_+$ and $\mathbf{L}_-$. In total, given a state $s$, the MLP computes $\mathbf{m}$, $\boldsymbol{\Sigma}$, $V$, $K$, $\mathbf{L}_+$ and $\mathbf{L}_-$. From Eq.7, for any action $a$ the advantage is uniquely defined: $A^{\mathbf{w}}(s, a) := f^{\mathbf{w}}(s, a) - \mathbb{E}_{a' \sim \pi}\left[f^{\mathbf{w}}(s, a')\right]$. Like the exact on-policy advantage $A^{\pi}$, $A^{\mathbf{w}}$ has by design expectation zero under the policy. The parameterization coefficients $K$ and $\mathbf{L}$ are updated to minimize the L2 error from $\hat{A}_t^{\text{ret}}$:

$$\hat{\mathbf{g}}_t^A(\mathbf{w}) = \rho_t\left[\hat{A}_t^{\text{ret}} - A^{\mathbf{w}}(s_t, a_t)\right]\nabla_{\{K,\,\mathbf{L}\}}A^{\mathbf{w}}(s_t, a_t) \tag{8}$$

Here, $\rho_t$ reduces the weight of estimation errors for unlikely actions, where $A^{\mathbf{w}}$ is expected to be less accurate. To ensure that $\boldsymbol{\Sigma}$, $K$, $\mathbf{L}_+$ and $\mathbf{L}_-$ are positive definite, the respective outputs are mapped onto $\mathbb{R}^+$ by a Softplus rectifier. The analytical derivation of $\mathbb{E}_{a' \sim \pi}\left[f^{\mathbf{w}}(s, a')\right]$ can be found in App. A. Collectively, $\hat{\mathbf{g}}_t^{\pi}(\mathbf{w})$, $\hat{\mathbf{g}}_t^V(\mathbf{w})$, and $\hat{\mathbf{g}}_t^A(\mathbf{w})$ form a vector $\hat{\mathbf{g}}_t^{\text{AC}}(\mathbf{w})$ with the same size as the MLP output. $\hat{\mathbf{g}}_t^{\text{AC}}(\mathbf{w})$ is weighted with the ReF-ER penalization (Eq. 3) and then back-propagated to the MLP.

In order to estimate $\hat{Q}_t^{\text{ret}}$ for a sampled time step $t$, Retrace (Eq. 2) requires $V^{\mathbf{w}}$ and $Q^{\mathbf{w}}$ for all future steps in sample $t$'s episode. These are naturally computed when training from batches of episodes (as in ACER) rather than time steps. However, the information contained in consecutive steps is highly correlated, worsening the quality of the gradient estimate, and episodes can be composed of thousands of time steps, increasing the computational cost. In order to efficiently train from uncorrelated time steps, RACER stores for each sample in the RM the most recently computed estimations of $V^{\mathbf{w}}(s_t)$, $A^{\mathbf{w}}(s_t, a_t)$, $\rho_t$ and $Q_t^{\text{ret}}$. When a batch of time steps is sampled, the stored $Q_t^{\text{ret}}$ are used to compute the gradient update. At the same time, the stored $V^{\mathbf{w}}(s_t)$, $A^{\mathbf{w}}(s_t, a_t)$ and $\rho_t$ are updated with the current NN outputs and used to correct $Q^{\text{ret}}$ for all prior time-steps in the respective episodes with Eq. 2. A description of all remaining implementation details is provided in App. D.

## 5 RELATED WORK

The rules that determine which samples are kept in the RM and how they are used for training can be designed to address several objectives. For example, properly designed ER may prove necessary to

prevent lifelong learning agents from forgetting previously mastered tasks (Isele & Cosgun, 2018). *Prioritized Experience Replay* (Schaul et al., 2015b) (`PER`) improves the performance of `DQN` (Mnih et al., 2015) by biasing sampling in favor of experiences that cause large temporal-difference (TD) errors. TD errors may signal rare events that would convey useful information to the learner. ER can be used to train transition models in planning-based RL (Pan et al., 2018), or to train off-policy learners on auxiliary tasks (Schaul et al., 2015a; Jaderberg et al., 2017) helping to shape the network features. When rewards are very sparse, RL agents can be trained to repeat previous outcomes (Andrychowicz et al., 2017) or to reproduce successful states or episodes (Oh et al., 2018; Goyal et al., 2018).

de Bruin et al. (2015) proposes a modification to ER that increases the diversity of behaviors contained in the RM, which is the opposite of what `ReF-ER` achieves. The ideas proposed by de Bruin et al. (2015) cannot readily be applied to complex tasks (they consider a low-dimensional problem which can be learned with the RM holding a handful of episodes) and the authors concede that their method is not suitable when the policy is advanced for many iterations. For these reasons, we compare `ReF-ER` only to `PER` and conventional ER. We assume that if increasing the diversity of experiences in the RM were beneficial to off-policy RL then either `PER` or ER would outperform `ReF-ER`.

`ReF-ER` is inspired by the techniques developed for on-policy RL to bound policy updates of `PPO` (Schulman et al., 2017). Rule 1 of `ReF-ER` is similar to the clipped objective function of `PPO` (gradients are zero if $\rho$ lies outside of some range). However, Rule 1 is not affected by the sign of the advantage estimate and clips both policy and value gradients. Like Rule 3, one variant of `PPO` penalizes $D_{KL}(\mu_t||\pi^{\mathtt{w}})$ (also Schulman et al. (2015) and Wang et al. (2017) employ trust-region schemes in the on- and off-policy setting respectively). While `PPO` picks one of the two techniques, in `ReF-ER` Rules 1 and 3 complement each other and can be applied to most off-policy RL methods with parametric policies. In the ER setting, samples remain in the RM over many iterations. If only Rule 1 is included, any PG could push the sample's $\rho$ outside of the clipping range without means to recover, leading to zero-valued gradients. Including only Rule 3 would not reduce the number of hyper-parameters (a target $D_{KL}$ would be needed), and would not prevent unbounded $\rho$.

The first method to combine ER and PG was the *Off-Policy Actor Critic* (Degris et al., 2012). Further contributions were introduced with `ACER` (Wang et al., 2017), such as extension to NN, estimation of on-policy returns with Retrace, and the *variance truncation and bias correction trick* (TBC). In the continuous-action domain, the proposed `RACER` offers several advantages over `ACER`: (a) reduced complexity by employing a single NN and closed-form $A^{\mathtt{w}}$ (continuous-`ACER` uses 9 NN evaluations per gradient). (b) relies on `ReF-ER` rather than constraining policy updates around a target network. (c) samples time steps rather than episodes (which may consist of thousands of steps). (d) uses the original Retrace estimator, which has better convergence guarantees than the value-estimators used by continuous-`ACER`. Because of (a) and (c), `RACER` is two orders of magnitude faster than `ACER`.

## 6 RESULTS

In this section we couple `ReF-ER`, conventional ER and `PER` with one method from each of the three main classes of deep continuous-action RL algorithms: `DDPG`, `NAF`, and `RACER`. The performance of each combination of algorithms is measured on the MuJoCo (Todorov et al., 2012) tasks of OpenAI Gym (Brockman et al., 2016) by plotting the mean cumulative reward $R = \sum_t r_t$. Each plot tracks the average $R$ among all episodes entering the RM within intervals of $2 \cdot 10^5$ time steps and averaged again among five differently seeded training trials. In the appendix we include contours of the $20^{th}$ to $80^{th}$ percentiles of $R$ obtained by `DDPG` and `RACER` on the OpenAI Gym and on the DeepMind Control Suite (Tassa et al., 2018). The code to reproduce all present results is available on GitHub.[1]

### 6.1 RESULTS FOR DDPG

`DDPG` (Lillicrap et al., 2016) trains two networks by ER. The value network outputs $Q^{\mathtt{w}'}(s, a)$ and is trained to minimize the L2 distance from the Q-learning target (Sec. 2). The policy network outputs a deterministic policy $\mathbf{m}^{\mathtt{w}}(s)$ and is trained with the DPG (Silver et al., 2014):

$$\hat{g}_t^{\text{DPG}}(\mathtt{w}) = \nabla_{\mathtt{w}}\mathbf{m}^{\mathtt{w}}(s) \left. \nabla_a Q^{\mathtt{w}'}(s, a)\right|_{a=\mathbf{m}^{\mathtt{w}}(s)} \tag{9}$$

---

[1] The repository is hidden to maintain anonymity during the review process.

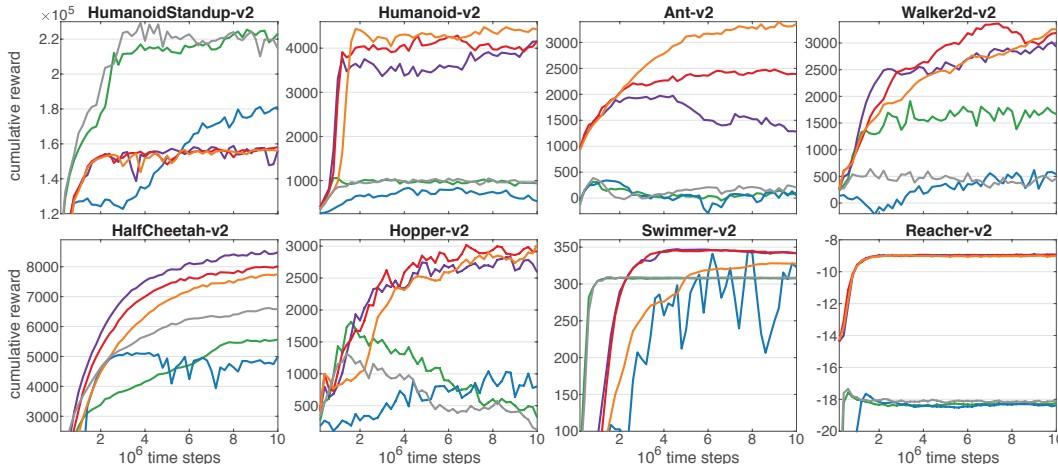

Figure 2: Cumulative rewards on OpenAI MuJoCo tasks for `DDPG` (green line), `DDPG` with rank-based `PER` (gray line), `u-DDPG` with regular ER (blue), and `ReF-ER` with $C = 8$ (purple), $C = 4$ (red), $C = 2$ (orange). Implementation details in App. D.

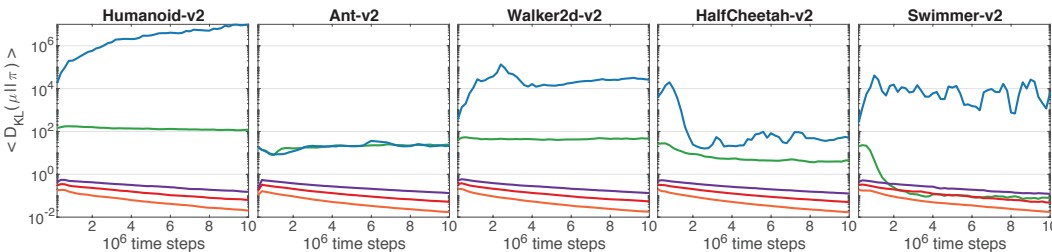

Figure 3: Average Kullback-Leibler divergence between the policy $\pi^{\mathtt{w}} = m^{\mathtt{w}} + \mathcal{N}(\mathbf{0}, \sigma^2 \mathbf{I})$ and the behaviors $\mu_t$ in the RM during training for `DDPG` (green line), `u-DDPG` with regular ER (blue), and `ReF-ER` with $C = 8$ (purple), $C = 4$ (red), $C = 2$ (orange).

Noise is added to the deterministic policy $\pi^{\mathtt{w}} = \mathbf{m}^{\mathtt{w}} + \mathcal{N}(\mathbf{0}, \sigma^2 \mathbf{I})$ for exploration. We consider two variants of `DDPG`: one with $\mathbf{m}^{\mathtt{w}}$ bounded to the unit box $[-1, 1]^{d_A}$ (as in the original) and one without bounds (referred to as `u-DDPG`, implementation details can be found in App. D). Bounding the policy may lead to lower returns in the OpenAI MuJoCo benchmarks, which are defined for unbounded actions. We note that, without measures to constrain policy updates or without careful tuning of the hyper-parameters (we found the critic's weight decay and temporally-correlated action noise to be necessary), `u-DDPG` is unstable. The returns for `u-DDPG` can fall to large negative values, especially in tasks that include a control-cost penalty in the reward such as Ant. This behavior is explained by the value network not having learned local maxima with respect to the action (Silver et al., 2014).

By replacing ER with `ReF-ER` we can stabilize `u-DDPG` and greatly improve its performance, especially for tasks with complex dynamics such as Humanoid or Ant. The hyper-parameter $c_{\max}$ determines how much the policy is allowed to change from the behaviors in the RM. By annealing $c_{\max}$ we can allow faster improvements at the beginning of training, when an inaccurate policy gradient might be sufficient to estimate a good direction for the update. Conversely, during the later stages of training, precise updates can be computed from almost on-policy samples. We anneal $c_{\max}$ and the learning rate according to:

$$c_{\max}(k) = 1 + C/(1 + 5e{-}7 \cdot k), \quad \eta(k) = \eta/(1 + 5e{-}7 \cdot k) \tag{10}$$

Here $k$ is the gradient step index, $\eta$ is the initial learning rate ($10^{-4}$ and $10^{-5}$ for the value and policy networks respectively, as when using regular ER). We found that annealing $\eta$ worsened the instability of `u-DDPG` with regular ER. Lower values of $C$ reduce the speed of policy improvements, but after $10^7$ time steps $C = 2$ achieves the best performance in most tasks. In Fig. 3 we report for a subset of problems the average $D_{KL}(\mu_t || \pi^{\mathtt{w}})$. $D_{KL}$ decreases for lower values of $C$ and further

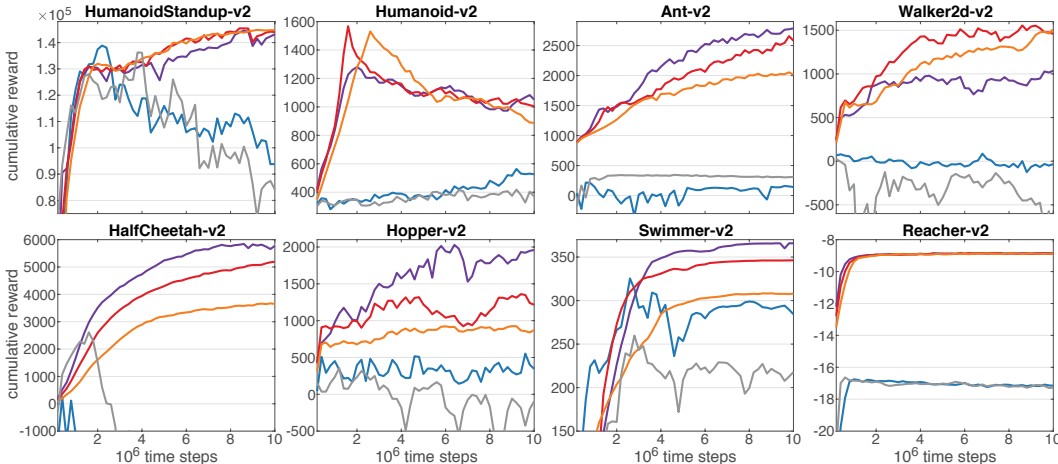

Figure 4: Cumulative rewards on the OpenAI MuJoCo tasks for `NAF` (blue line), `NAF` with rank-based `PER` (gray line), `NAF` with `ReF-ER` and $C = 8$ (purple), $C = 4$ (red), $C = 2$ (orange).

decreases during training due to the annealing process. With regular ER, even after lowering $\eta$ by one order of magnitude from that of the original paper, the distance between $\pi$ and $\mu$ may span the entire action space. In fact, in most tasks shown in Fig. 3 the $D_{KL}(\mu_t || \pi^{\mathtt{w}})$ of `DDPG` is of similar order of magnitude as its maximum $\frac{d_A}{2}(2/\sigma)^2$ (for example, since $\sigma = 0.2$, the maximum $D_{KL}$ is 850 for Humanoid and 300 for Walker2d and it oscillates during training around 100 and 50 respectively).

`ReF-ER` maintains a RM of mostly near-policy samples, providing the value-network with multiple examples of trajectories that are possible with the current policy. This focuses the predictive capabilities of the value-network, enabling it to extrapolate the effect of a marginal change of action on the expected returns, and therefore increasing the accuracy of the DPG. Any misstep of the DPG is weighted with a penalization term that attracts the policy towards past behaviors. This allows time for the learner to gather experiences with the new policy, improve the value-network, and correct the misstep. This reasoning is almost diametrically opposed to that behind `PER`. In `PER` observations that are associated with larger TD errors are sampled more frequently. In the continuous-action setting, however, TD errors may be result from actions that are farther from the current policy. Therefore, precisely estimating their value might not help the value network in yielding an accurate estimate of the DPG. We obtained better results with the rank-based variant of `PER`, which has similar performance to that of `DDPG` with ER. The main benefits over `ReF-ER` arise in tasks that require more exploration, such as Swimmer and HumanoidStandup.

## 6.2 RESULTS FOR `NAF`

Normalized Advantage Functions (`NAF`) is the state-of-the-art of Q-learning based algorithms for continuous-action problems. It employs a quadratic-form approximation of the advantage $A^{\mathtt{w}}$, analogous to the one employed by `RACER`:

$$A^{\mathtt{w}}_{\mathrm{NAF}}(s, a) = -\left[a - \mathbf{m}^{\mathtt{w}}(s)\right]^{\mathsf{T}} \mathbf{L}^{\mathtt{w}}_{\mathrm{Q}}(s) \left[\mathbf{L}^{\mathtt{w}}_{\mathrm{Q}}(s)\right]^{\mathsf{T}} \left[a - \mathbf{m}^{\mathtt{w}}(s)\right] \tag{11}$$

`NAF` trains $A^{\mathtt{w}}$ with the Q-learning target (Sec. 2) and uses the location $\mathbf{m}^{\mathtt{w}}$ of its maximum as the mean of the policy, with added Gaussian noise for exploration. When the actual $A^{\pi}$ is not well approximated by a quadratic (e.g. when the return landscape is multi-modal), `NAF` may fail to choose good actions. In contrast, `RACER` updates $\pi^{\mathtt{w}}$ with the off-PG, which is independent of the error in approximating $A^{\pi}(s_t, a_t)$ (but depends on the error at $t+1$). Figure 4 shows how `NAF` is affected by the choice of ER algorithm. While Q-learning based methods are thought to be less sensitive than PG-based methods to the dissimilarity between policy and stored behaviors owing to the bootstrapped Q-learning target, `NAF` benefits from `REF-ER`. This is because $A^{\pi}$ is likely to be approximated well by a quadratic in a small neighborhood near its local maxima. `ReF-ER` constrains learning from actions within this neighborhood and prevents large TD errors from disrupting the locally-accurate approximation of $A^{\mathtt{w}}$. This intuition is supported by observing that rank-based `PER` (the better performing variant of `PER` also in this case), often worsens the performance of `NAF`. `PER`

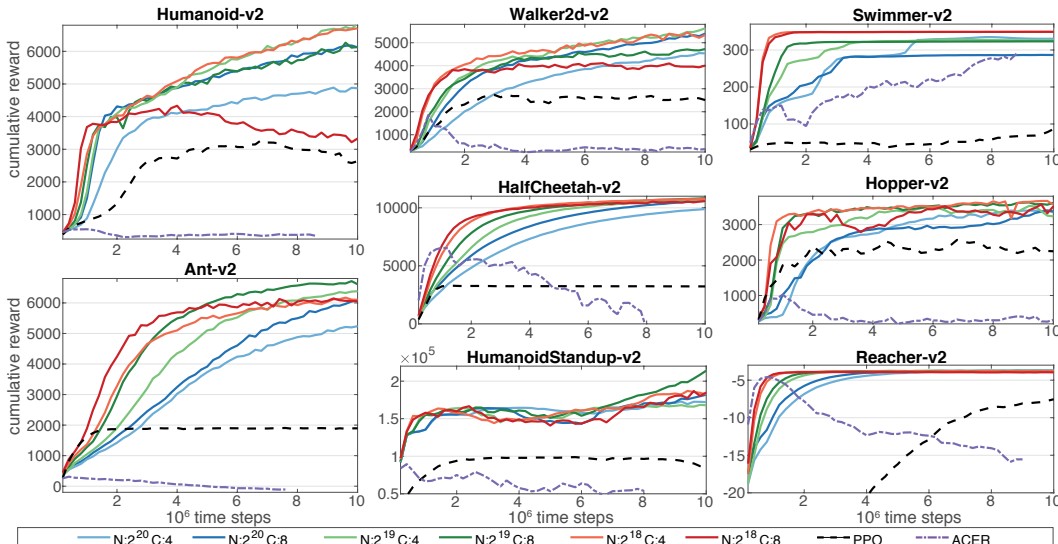

Figure 5: Average cumulative rewards on MuJoCo OpenAI Gym tasks obtained with PPO (dashed black lines), ACER (dash-dot purple lines) and with RACER by independently varying the two main hyper-parameters of ReF-ER: the RM size $N$ and $C$ (colored lines).

aims at biasing sampling in favor of larger TD errors, which are more likely to be farther from $\mathbf{m}(s)$ (note that $f_{\mathrm{NAF}}^{\mathtt{w}}$ is unbounded), and their accurate prediction might not help the learner in fine-tuning the policy by improving a local approximation of the advantage.

## 6.3 RESULTS FOR RACER

Here we compare RACER to ACER and PPO, an algorithm that owing to its simplicity and good performance on MuJoCo tasks is often used as baseline. We omit results from coupling RACER with ER or PER as it yields large negative $R$ values. In fact, without ReF-ER, the unbounded importance weights cause off-PG estimates to diverge, disrupting prior learning progress. Similarly to ReF-ER, ACER's techniques (Sec. 4) guard against the numerical instability of the off-PG. However, bounding policy updates to a target-network does not ensure similarity between $\pi^{\mathtt{w}}$ and RM behaviors (as shown in Fig. 7). In fact, when using deep approximators, simply enforcing slow parameter updates does not guarantee small changes in the output. As $D_{KL}(\mu_t \parallel \pi^{\mathtt{w}})$ grows, off-PG estimates may become inaccurate, causing ACER to be outperformed by RACER and PPO. We note that, due to ACER's cost, we could only optimize some of its hyper-parameters relative to the original paper (see App. D). We do not exclude that ACER could have performed better after more extensive tuning.

The two hyper-parameters that most strongly affect the performance of RACER are the RM size $N$ and $c_{\mathrm{max}}$, annealed with Eq. 10 (for a discussion of the other parameters see App. B). RACER uses the Retrace estimator $Q^{\mathrm{ret}}$ which is expected to converge to the on-policy $Q^{\pi}$ (Munos et al., 2016). Because $\pi^{\mathtt{w}}$ is gradually changing during training, it is crucial to maximize the convergence speed of Retrace to obtain accurate estimates of the off-PG. Large values of $c_{\mathrm{max}}$ increase the variance of the policy gradient and increase the amount of "trace-cutting" as the importance weights are allowed to diverge from 1. The cumulative rewards reported in Fig. 5 show that "stable" tasks, where the agent's success is less predicated on avoiding mistakes that would cause it to trip, are more tolerant to high values of $c_{\mathrm{max}}$ (e.g. HalfCheetah). In most other tasks, especially those that require precise controls (e.g. Walker), the best results are obtained for values of $c_{\mathrm{max}}$ that strike a balance between strict penalty terms, which slow down policy improvements, and high-variance gradient estimates. The RM size $N$ has a parallel effect. A small RM may not contain enough diversity of samples for the learner to accurately estimate the gradients. Conversely, a large RM is composed of episodes obtained with increasingly older versions of $\pi^{\mathtt{w}}$. In this case, the penalty terms required to preserve a sufficient fraction of near-policy samples may prevent the policy from improving. A discussion of all the secondary hyper-parameters and the choice of function parameterizing $A^{\mathtt{w}}$ can be found in App. B. For most combinations of hyper-parameters and tasks presented in this section, RACER

outperforms the best result from `DDPG` (Sec. 6.1), `PPO`, and is competitive with the best results found in the published literature, which to our knowledge were achieved by the on-policy algorithms `TRPO` (Schulman et al., 2015) and Policy Search with Natural Gradient (Rajeswaran et al., 2017).

# 7 CONCLUSION

Many RL algorithms update a policy $\pi^w$ from past experiences collected with off-policy behaviors $\mu$. We present evidence that off-policy continuous-action deep RL methods benefit from actively maintaining similarity between policy and replay behaviors. We propose a novel ER algorithm (`ReF-ER`) that extends these benefits to off-policy PG, deterministic PG and Q-learning methods. `ReF-ER` characterizes past behaviors either as "near-policy" or "far-policy" by the deviation from one of the importance weight $\rho = \pi^w(a|s)/\mu(a|s)$. `ReF-ER` consists of: 1) Computing gradients only from near-policy experiences. 2) Forgetting far-policy samples when new observations are obtained from the environment. 3) Regulating the pace at which $\pi^w$ is allowed to deviate from $\mu$ through penalty terms that reduce $D_{KL}(\mu||\pi^w)$. This allows time for the learner to gather experiences with the new policy, improve the value estimators, and increase the accuracy of the next steps. The extension of `ReF-ER` to discrete-action methods is subject of ongoing work.

`ReF-ER` is combined with a novel method based on the off-policy PG (`RACER`). `RACER` uses a closed-form parameterization of the advantage and it estimates on-policy returns with a bootstrapped formulation of Retrace. These innovations greatly increase the computational efficiency of `RACER` compared to similar methods. The application of `ReF-ER` and `RACER` to OpenAI Gym benchmarks produces state-of-the-art results while being robust to large variations in their hyper-parameters.

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

## A EXPECTATIONS OF THE PARAMETERIZED ADVANTAGE FUNCTIONS

We considered three parameterizations for the action advantage:

$$f_Q^{\mathtt{w}}(s,a) = -\tfrac{1}{2} \left[ a - \mathbf{m}(s) \right]^{\mathsf{T}} \mathbf{L}_Q(s) \mathbf{L}_Q^{\mathsf{T}}(s) \left[ a - \mathbf{m}(s) \right] \tag{12}$$

$$f_{\mathrm{SG}}^{\mathtt{w}}(s,a) = K(s) \, \exp \left\{ -\tfrac{1}{2} \left[ a - \mathbf{m}(s) \right]^{\mathsf{T}} \mathbf{L}_{\mathrm{SG}}^{-1}(s) \left[ a - \mathbf{m}(s) \right] \right\} \tag{13}$$

$$f_{\mathrm{DG}}^{\mathtt{w}}(s,a) = K(s) \, \exp \left\{ -\tfrac{1}{2} \mathbf{a}_+^{\mathsf{T}} \, \mathbf{L}_+^{-1}(s) \, \mathbf{a}_+ - \tfrac{1}{2} \mathbf{a}_-^{\mathsf{T}} \, \mathbf{L}_-^{-1}(s) \, \mathbf{a}_- \right\} \tag{14}$$

We recall $\mathbf{a}_- = \min \left[ a - \mathbf{m}(s), \mathbf{0} \right]$ and $\mathbf{a}_+ = \max \left[ a - \mathbf{m}(s), \mathbf{0} \right]$. The first parameterization $f_Q^{\mathtt{w}}$ is identical to the one employed by NAF (Gu et al., 2016), and requires training $(d_A^2 + d_A)/2$ coefficients of the lower triangular matrix $\mathbf{L}_Q$. Its quadratic complexity makes the choice of $f_Q^{\mathtt{w}}$ unfavorable for high-dimensional action spaces (e.g. it requires 153 parameters for the 17-dimensional Humanoid tasks of OpenAI Gym, against the 35 of $f_{\mathrm{DG}}^{\mathtt{w}}$). In order to preserve bijection between $\mathbf{L}_Q$ and $\mathbf{L}_Q \mathbf{L}_Q^{\mathsf{T}}$, the diagonal terms are mapped to $\mathcal{R}^+$ with a Softplus function. The expectation under a Gaussian policy can be computed as (Petersen et al., 2008):

$$\mathbb{E}_{a' \sim \pi} \left[ f_Q^{\mathtt{w}}(s,a') \right] = \mathrm{Tr} \left[ \mathbf{L}_Q(s) \mathbf{L}_Q^{\mathsf{T}}(s) \Sigma(s) \right] \tag{15}$$

Here Tr denotes the trace of a matrix.

The second parameterization requires training the $1 + d_A$ coefficients for $K(s)$ and the diagonal matrix $\mathbf{L}(s)$. The expectation can be easily derived from the properties of products of Gaussian densities:

$$\mathbb{E}_{a' \sim \pi} \left[ f_{\mathrm{SG}}^{\mathtt{w}}(s,a') \right] = K(s) \sqrt{ \frac{ |\mathbf{L}(s)| }{ |\mathbf{L}(s) + \boldsymbol{\Sigma}(s)| } } \tag{16}$$

Here $| \cdot |$ denotes a determinant.

To derive the expectation of the third parameterization, which was used for most results in the paper, we recall that the expectation of a product of independent variables is the product of the expectations. For one component $i$ of the action vector:

$$\mathbb{E}_{a' \sim \pi} \left[ e^{ -\frac{1}{2} u_{+,i}^{\mathsf{T}} \, L_{+,i}^{-1}(s) \, u_{+,i} - \frac{1}{2} u_{-,i}^{\mathsf{T}} \, L_{-,i}^{-1}(s) \, u_{-,i} } \right] = \frac{ \sqrt{ \frac{L_{+,i}(s)}{L_{+,i}(s)+\Sigma_i(s)} } + \sqrt{ \frac{L_{-,i}(s)}{L_{-,i}(s)+\Sigma_i(s)} } }{2} \tag{17}$$

Here we exploited the symmetry of the Gaussian policy around the mean. Since $\boldsymbol{\Sigma}$, $\mathbf{L}_+$, and $\mathbf{L}_-$ are all diagonal, we can compute the expectation:

$$\mathbb{E}_{a' \sim \pi} \left[ f_{\mathrm{DG}}^{\mathtt{w}}(s,a') \right] = K(s) \prod_{i=1}^{d_A} \frac{ \sqrt{ \frac{L_{+,i}(s)}{L_{+,i}(s)+\Sigma_i(s)} } + \sqrt{ \frac{L_{-,i}(s)}{L_{-,i}(s)+\Sigma_i(s)} } }{2} \tag{18}$$

Finally, we note that all these parameterizations are differentiable.

## B SENSITIVITY TO HYPER-PARAMETERS

Figure 6 shows the robustness of RACER to various hyper-parameters and to the choice of the advantage parameterizations introduced in App. A. Moreover, we can bypass the advantage approximator (i.e. $A^{\mathtt{w}} := 0$) and rely exclusively on the state value approximator. We recall the on-policy value estimator obtained with "variance truncation and bias correction" (TBC) (Wang et al., 2017) (Sec. 4):

$$\hat{V}_t^{\mathrm{tbc}} = V^{\mathtt{w}}(s_t) + \min\{1, \rho_t\}(\hat{Q}_t^{\mathrm{ret}} - Q^{\mathtt{w}}(s_t, a_t)) \tag{19}$$

From Eq. 2 and 19 we obtain $\hat{Q}^{\mathrm{ret}}(s_t, a_t) = r_{t+1} + \gamma \hat{V}^{\mathrm{tbc}}(s_{t+1})$. By neglecting $A^{\mathtt{w}}$, from this relation and Eq. 19, we obtain a recursive Retrace-based estimator for the on-policy state value that depends on $V^{\mathtt{w}}$ alone:

$$\hat{V}_t^{\mathrm{tbc}} = V^{\mathtt{w}}(s_t) + \min\left\{1, \, \rho(s_t, a_t)\right\} \left[ r_{t+1} + \gamma \hat{V}^{\mathrm{tbc}}(s_{t+1}) - V^{\mathtt{w}}(s_t) \right] \tag{20}$$

This target is equivalent to the recently proposed V-trace estimator (Espeholt et al., 2018) when all importance weights are clipped at 1, which was empirically found by the authors to be the best-performing solution. As expected, lacking a model for $A^{\mathtt{w}}(s, a)$, this architecture (denoted

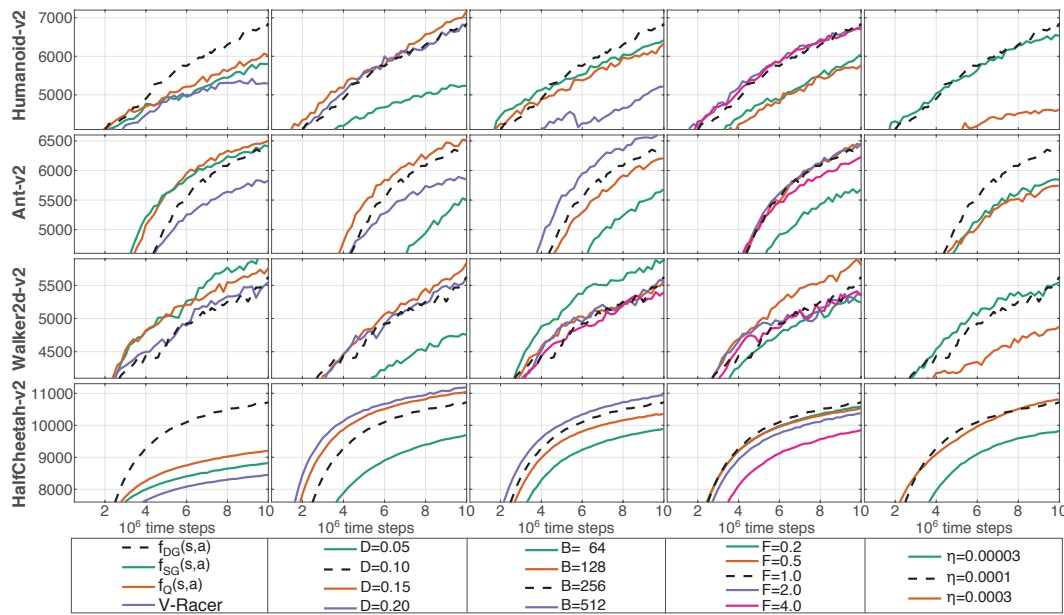

Figure 6: Mean cumulative rewards with `RACER` on OpenAI Gym tasks exploring the effect of the choice of advantage parameterization, `ReF-ER` tolerance factor $D$, mini-batch size $B$, number of time steps per gradient step $F$, and learning rate $\eta$. Dashed-black lines refer to the baseline parameters. The $y$-axes are magnified around the final returns to clarify often minor differences in performance.

as `V-RACER` in the first column of results of Fig. 6) yielded worse performance than the original `RACER`. However the difference is often minor and the increased simplicity of this architecture might justify its adoption for some problems.

The tolerance $D$ for far-policy samples in the RM has a similar effect as $c_{\max}$: low values tend to delay learning while high values reduce the fraction of the RM that is used to compute updates and may decrease the accuracy of gradient estimates. Increasing the number $F$ of time steps per gradient step could either cause a rightward shift of the expected returns because the learner computes fewer updates for the same budget of observations, or it could aid the learner by providing more on-policy samples. The actual outcomes depend on the problem: tasks with simpler dynamics (e.g. HalfCheetah) can be learned more quickly by performing more gradient steps, while problems with more complex dynamics (e.g. Ant, Humanoid) benefit from more on-policy samples. The batch-size $B$ and learning rate $\eta$ have a minor effect on performance.

## C  STATE, ACTION AND REWARD PREPROCESSING

Several authors have employed state (Henderson et al., 2017) and reward (Duan et al., 2016) (Gu et al., 2017) rescaling to improve the learning results. This technique has not systematically been studied but reflects a deeper challenge of RL. The success of any RL algorithm is intrinsically linked to the design of the control problem. For example, the stability of `DDPG` is affected by the L2 weight decay of the value-network. Depending on the numerical values of the distribution of rewards provided by the environment and the choice of weight decay coefficient, the L2 penalization can be either negligible or dominate the Bellman error. Similarly, the distribution of values describing the state variables can increase the challenge of learning by gradient descent.

We propose partially addressing these issues by rescaling both rewards and state vectors depending on the the experiences contained in the RM. At the beginning of training we prepare the RM by collecting $N_{\text{start}}$ observations and then we compute:

$$\mu_s = \frac{1}{n_{\text{obs}}} \sum_{t=0}^{n_{\text{obs}}} s_t \tag{21}$$

$$\sigma_s = \sqrt{\frac{1}{n_{\text{obs}}} \sum_{t=0}^{n_{\text{obs}}} \left(s_t - \mu_s\right)^2} \tag{22}$$

Throughout training, $\mu_s$ and $\sigma_s$ are used to standardize all state vectors $\hat{s}_t = (s_t - \mu_s)/(\sigma_s + \epsilon)$ before feeding them to the network approximators. Moreover, every 1000 steps, chosen as the smallest power of ten that did not affect the run time, we loop over the $n_{\text{obs}}$ observations stored in the RM to compute:

$$\sigma_r \leftarrow \sqrt{\frac{1}{n_{\text{obs}}} \sum_{t=0}^{n_{\text{obs}}} \left( r_{t+1} \right)^2} \tag{23}$$

This value is used to scale the rewards $\hat{r}_t = r_t/(\sigma_r + \epsilon)$ used by the Q-learning target of `DDPG` and the Retrace algorithm for `RACER`. We use $\epsilon = 10^{-7}$ to ensure numerical stability.

The actions sampled by the learner may need to be rescaled or bounded to some interval depending on the environment. For the OpenAI Gym tasks this amounts to a linear scaling $a' = a\,(\texttt{upper\_value} - \texttt{lower\_value})/2$, where the values specified by the Gym library are $\pm 0.4$ for the Humanoid tasks, $\pm 8$ for the Pendulum tasks, and $\pm 1$ for all others. The tasks of DeepMind Control Suite are defined with all actions bounded do the interval $[-1, 1]^{d_A}$. In this case we can learn Gaussian policies, as if the action space were unbounded, and then map the actions sent to the environment with an hyperbolic tangent: $a'_t = \tanh a_t$. This approach, however, might prevent efficiently learning policies that behave like bang-bang controllers. Close to the bounds of the control space, actions that should be similar may be mapped to distant positions in the unbounded action space. Therefore, learning bang-bang controls is more likely to be hindered by the $D_{KL}$ penalties of `ReF-ER`. This issue may explain some poor results on the DeepMind Control Suite shown in Fig. 10 and 11. These results show that successfully tackling control problems with bounded action spaces may require policies parameterized as Beta distribution (Chou et al., 2017).

## D    IMPLEMENTATION AND NETWORK ARCHITECTURE DETAILS

We implemented all presented learning algorithms within `smarties`,[2] our open source C++ RL framework, and optimized for high CPU-level efficiency through fine-grained multi-threading, strict control of cache-locality, and computation-communication overlap. On every step, we asynchronously obtain on-policy data by sampling the environment with $\pi$, which advances the index $t$ of observed time steps, and we compute updates by sampling from the Replay Memory (RM), which advances the index $k$ of gradient steps (Fig. 1a). During training, the ratio of time and update steps is usually equal to a constant: $t/k = F$. This parameter affects the data efficiency of the algorithm; by lowering $F$ each sample is used more times to improve the policy before being replaced by newer samples. Upon completion of all tasks, we apply the gradient update and proceed to the next step. The pseudo-codes in App. E neglect parallelization details as they do not affect execution.

In order to evaluate all algorithms on equal footing, we use the same baseline network architecture for `RACER`, `DDPG` and `NAF`, consisting of an MLP with two hidden layers of 128 units each. For the sake of computational efficiency, we employed Softsign activation functions. The weights of the hidden layers are initialized according to $\mathcal{U}\left[-6/\sqrt{f_i + f_o},\ 6/\sqrt{f_i + f_o}\right]$, where $f_i$ and $f_o$ are respectively the layer's fan-in and fan-out (Glorot & Bengio, 2010). The weights of the linear output layer are initialized from the distribution $\mathcal{U}\left[-0.1/\sqrt{f_i},\ 0.1/\sqrt{f_i}\right]$, such that the MLP has near-zero outputs at the beginning of training. When sampling the components of the action vectors, the policies are treated as truncated normal distributions with symmetric bounds at three standard deviations from the mean. Finally, we optimize the network weights with the Adam algorithm (Kingma & Ba, 2015).

**RACER** We note that the values of the diagonal covariance matrix are shared among all states and initialized to $\mathbf{\Sigma} = 0.2\mathbf{I}$. The remaining hyper-parameters of `RACER` are listed in table 1.

**DDPG** In its original formulation, `DDPG` transforms the policy-network's output onto the bounded interval $[-1, 1]^{d_A}$ with an hyperbolic tangent function. We found that bounding the action space may limit the performance of `DDPG` with the OpenAI MuJoCo tasks. In order to stabilize the unbounded-action version of `DDPG` (`u-DDPG`) we set the learning rate for the policy-network to $1 \cdot 10^{-5}$ and that of the value-network to $1 \cdot 10^{-4}$ with L2 weight decay coefficient of $1 \cdot 10^{-4}$. These changes lead to better performance also with the bounded-action `DDPG` and therefore were used for all numerical experiments. The RM is set to contain $N = 2^{19}$ observations and we follow Henderson et al. (2017) for the remaining hyper-parameters: mini-batches of $B = 128$ samples, $\gamma = 0.995$, soft target network

---

[2] The repository is hidden to maintain anonymity during the review process.

Table 1: RACER architecture's baseline hyper-parameters.

| Short-hand | Description | Baseline value |
|---|---|---|
| $C$ | Annealing parameter of $c_{\max}$. | 4 |
| $N$ | Size of the Replay Memory. | $2^{19}$ |
| $N_{\text{start}}$ | Number of samples gathered before training starts. | $2^{18}$ |
| $D$ | Fraction of far-policy samples allowed in the RM. | 0.1 |
| $B$ | Mini-batch size. | 256 |
| $F$ | Ratio between observed time steps and gradient steps. | 1 |
| $\eta$ | Learning rate. | $10^{-4}$ |
| $\gamma$ | Discount factor. | 0.995 |

update coefficient 0.01. We note that while DDPG is the only algorithm employing two networks, choosing half the batch-size as RACER and NAF makes the compute cost roughly equal among the three methods. Finally, when using ReF-ER we add exploratory Gaussian noise to the deterministic policy: $\pi^{\mathbf{w}'}=\mathbf{m}^{\mathbf{w}'}+\mathcal{N}(\mathbf{0},\sigma^2\mathbf{I})$ with $\sigma=0.2$. When performing regular ER or PER we sample the exploratory noise from an Ornstein–Uhlenbeck process with $\sigma=0.2$ and $\theta=0.15$.

**NAF** We use the same baseline MLP architecture and learning rate $\eta = 10^{-4}$, batch-size $B = 256$, discount $\gamma = 0.995$, RM size $N = 2^{19}$, and soft target network update coefficient 0.01. Gaussian noise is added to the deterministic policy $\pi^{\mathbf{w}'}=\mathbf{m}^{\mathbf{w}'}+\mathcal{N}(\mathbf{0},\sigma^2\mathbf{I})$ with $\sigma=0.2$.

**PPO** We tuned the hyper-parameters as Henderson et al. (2017): $\gamma=0.995$, GAE $\lambda=0.97$, policy clipping at $\Delta\rho_t=0.2$, and we alternate performing 2048 environment steps and 10 optimizer epochs with batch-size 64 on the obtained data. Both the policy- and the value-network are 2-layer MLPs with 64 units per layer. We further improved results by having separate learning rates ($10^{-4}$ for the policy and $3 \cdot 10^{-4}$ for the critic) with the same annealing as used in the other experiments.

**ACER** We kept most hyper-parameters as described in the original paper (Wang et al., 2017): the TBC clipping parameter is $c = 5$, the trust-region update parameter is $\delta = 1$, and five samples of the advantage network are used to compute $A^{\mathbf{w}}$ estimates under $\pi$. We use a RM of $1e5$ samples, each gradient is computed from 24 uniformly sampled episodes, and we perform one gradient step per environment step. Because here learning is not from pixels, each network (value, advantage, and policy) is an MLP with 2 layers and 64 units per layer (we tried training runs with 96 and 128 units per layer and did not observe improvements). Accordingly, we reduced the soft target-network update coefficient ($\alpha = 0.001$) and the learning rates for the value- ($\eta = 3 \cdot 10^{-4}$) and for the policy-network ($\eta = 1 \cdot 10^{-4}$). Despite these reduced learning rates, Fig. 7 shows that ACER's techniques do not prevent the policy from becoming disconnected from prior behaviors. We note that hyper-parameters are known to affect performance (Henderson et al., 2017) and we do not exclude that ACER could have performed better with more extensive tuning.

## E  PSEUDO-CODES

Remarks on algorithm 1: 1) It describes the general structure of the ER-based off-policy RL algorithms implemented for this work (i.e. RACER, DDPG, and NAF). 2) This algorithm can be adapted to conventional ER, PER (by modifying the sampling algorithm to compute the gradient estimates), or ReF-ER (by following Sec. 3)). 3) The algorithm requires 3 hyper-parameters: the ratio of time step to gradient steps $F$ (usually set to 1 as in DDPG), the maximal size of the RM $N$, and the minimal size of the RM before we begin gradient updates $N_{\text{start}}$.

Remarks on algorithm 2: 1) The reward for an episode's initial state, before having performed any action, is zero by definition. 2) The value $V^{\mathbf{w}}(s_t)$ for the last state of an episode is computed if the episode has been truncated due the task's time limits or is set to zero if $s_t$ is a terminal state. 3) Each time step we use the learner's updated policy network and we store $\mu_t = \{\mathbf{m}(s_t), \mathbf{\Sigma}(s_t)\}$.

Remarks on algorithm 3: 1) In order to compute the gradients $\hat{g}_{t_i}^{AC}(\mathbf{w})$, we rely on advantage estimates $Q_{t_i}^{\text{ret}}$ that were computed when subsequent time steps in $t_i$'s episode were previously drawn by ER. Not having to compute the quantities $A^{\mathbf{w}}$, $V^{\mathbf{w}}$, and $\rho$ for all following steps comes with clear computational

---

**Algorithm 1** Serial description of the master algorithm.

---

$t = 0$, $k = 0$,
Initialize an empty RM, network weights $\mathtt{w}$, and Adam's (Kingma & Ba, 2015) moments.
**while** $n_{\mathrm{obs}} < N_{\mathrm{start}}$ **do**
    Advance the environment according algorithm 2.
**end while**
Compute the initial statistics used to standardize the state vectors (App. C).
Compute the initial statistics used to rescale the rewards (App. C).
**while** $t < T_{\mathrm{max}}$ **do**
    **while** $t < F \cdot k$ **do**
        Advance the environment according to algorithm 2.
        **while** $n_{\mathrm{obs}} > N_{\mathrm{start}}$ **do**
            Remove an episode from the RM (either first in first out or as in Sec. 3).
        **end while**
        $t \leftarrow t + 1$
    **end while**
    Sample $B$ time steps from the RM to compute a gradient estimate (e.g. for RACER with algorithm 3).
    Perform the gradient step with the Adam algorithm.
    If applicable, update the ReF-ER penalization coefficient $\beta$.
    **if** modulo$(k, 1000)$ is 0 **then**
        Update the statistics used to rescale the rewards (App. C).
    **end if**
    $k \leftarrow k + 1$
**end while**

---

**Algorithm 2** Environment sampling

---

Observe $s_t$ and $r_t$.
**if** $s_t$ concludes an episode **then**
    Store data for $t$ into the RM: $\{s_t, r_t, V^{\mathtt{w}}(s_t)\}$
    Compute and store $Q^{\mathrm{ret}}$ for all steps of the episode
**else**
    Sample the current policy $a_t \sim \pi^{\mathtt{w}}(a|s_t) = \mu_t$
    Store data for $t$ into the RM: $\{s_t, r_t, a_t, \mu_t, V^{\mathtt{w}}(s_t), A^{\mathtt{w}}(s_t, a_t)\}$
    Advance the environment by performing $a_t$
**end if**

---

efficiency benefits, at the risk of employing an incorrect estimate for $Q_{t_i}^{\mathrm{ret}}$. In practice, we find that the Retrace values incur only minor changes between updates (even when large RM sizes decrease the

---

**Algorithm 3** RACER's gradient update

---

**for** mini-batch sample $i = 0$ to $B$ **do**
    Fetch all relevant information: $s_{t_i}$, $a_{t_i}$, $Q_{t_i}^{\mathrm{ret}}$, and $\mu_{t_i} = \{\mathbf{m}_{t_i}, \mathbf{\Sigma}_{t_i}\}$.
    Call the approximator to compute $\pi^{\mathtt{w}}$, $V^{\mathtt{w}}(s_{t_i})$, $A^{\mathtt{w}}(s_{t_i}, a_{t_i})$
    Update $Q^{\mathrm{ret}}$ for all prior steps in $t_i$'s episode with $V^{\mathtt{w}}(s_{t_i})$, $A^{\mathtt{w}}(s_{t_i}, a_{t_i})$
    Update the importance weight $\rho_{t_i} = \pi^{\mathtt{w}}(a_{t_i}|s_{t_i})/\mu_{t_i}(a_{t_i}|s_{t_i})$
    **if** $1/c_{\max} < \rho_{t_i} < c_{\max}$ **then**
        Compute $\hat{g}_{t_i}^{AC}(\mathtt{w})$ according to Sec. 4
    **else**
        $\hat{g}_{t_i}^{AC}(\mathtt{w}) = \mathbf{0}$
    **end if**
    ReF-ER penalization: $\hat{g}_{t_i}^{\mathrm{ReF\text{-}ER}}(\mathtt{w}) = \beta \hat{g}_{t_i}^{AC}(\mathtt{w}) - (1-\beta)\nabla D_{\mathrm{KL}}[\mu_{t_i}(\cdot|s_{t_i})||\pi^{\mathtt{w}}(\cdot|s_{t_i})]$
**end for**
Accumulate the gradient estimate over the mini-batch $\frac{1}{B}\sum_{i=0}^{B} \hat{g}_{t_i}^{\mathrm{ReF\text{-}ER}}(\mathtt{w})$

---

---

**Algorithm 4** `DDPG`'s gradient update with `ReF-ER`

---

**for** mini-batch sample $i = 0$ to $B$ **do**

    Fetch all relevant information: $s_{t_i}$, $a_{t_i}$, and $\mu_{t_i} = \{\mathbf{m}_{t_i}, \mathbf{\Sigma}_{t_i}\}$.

    The policy-network computes $\mathbf{m}^{\mathtt{w}}(s_{t_i})$ and the value-network computes $Q^{\mathtt{w}'}(s_{t_i}, a_{t_i})$.

    Define a stochastic policy with Gaussian exploration noise: $\pi^{\mathtt{w}}(a \mid s_{t_i}) = \mathbf{m}^{\mathtt{w}}(s_{t_i}) + \mathcal{N}$

    Update the importance weight $\rho_{t_i} = \pi^{\mathtt{w}}(a_{t_i}|s_{t_i})/\mu_{t_i}(a_{t_i}|s_{t_i})$

    **if** $1/c_{\max} < \rho_{t_i} < c_{\max}$ **then**

        Compute the policy at $t_i+1$ with the target-network: $\mathbf{m}^{\tilde{\mathtt{w}}}(s_{t_i+1})$

        Compute the Q-learning target: $\hat{q}_{t_i} = r_{t_i+1} + \gamma Q^{\tilde{\mathtt{w}}'}\left(s_{t_i+1}, \mathbf{m}^{\tilde{\mathtt{w}}}(s_{t_i+1})\right)$

        The gradient $g_{t_i}^{Q}(\mathtt{w}')$ of the value-network minimizes the squared distance from $\hat{q}_{t_i}$.

        The gradient $g_{t_i}^{\mathrm{DPG}}(\mathtt{w})$ of the policy-network is the deterministic PG (Eq. 9).

    **else**

        $g_{t_i}^{Q}(\mathtt{w}') = \mathbf{0}$, $g_{t_i}^{\mathrm{DPG}}(\mathtt{w}) = \mathbf{0}$

    **end if**

    `ReF-ER` penalization: $\hat{g}_{t_i}^{\mathrm{ReF\text{-}ER}}(\mathtt{w}) = \beta g_{t_i}^{\mathrm{DPG}}(\mathtt{w}) - (1-\beta)\nabla D_{\mathrm{KL}}[\mu_{t_i}(\cdot|s_{t_i})||\pi^{\mathtt{w}}(\cdot|s_{t_i})]$

**end for**

Accumulate the gradient estimates over the mini-batch for both networks.

Update the target policy- ($\tilde{\mathtt{w}} \leftarrow (1-\alpha)\tilde{\mathtt{w}} + \alpha\mathtt{w}$) and target value-networks ($\tilde{\mathtt{w}}' \leftarrow (1-\alpha)\tilde{\mathtt{w}}' + \alpha\mathtt{w}'$).

---

frequency of updates to the Retrace estimator) and that relying on previous estimates has no evident effect on performance. This could be attributed to the gradual policy changes enforced by `ReF-ER`. 2) With a little abuse of the notation, with $\pi$ (or $\mu$) we denote the statistics (mean, covariance) of the multivariate normal policy, with $\pi(a|s)$ we denote the probability of performing action $a$ given state $s$, and with $\pi(\cdot|s)$ we denote the probability density function over actions given state $s$.

Remarks on algorithm 4: 1) It assumes that weights and Adam are initialized for both policy-network and value-network. 2) The "target" weights are initialized as identical to the "trained" weights. 3) For the sake of brevity, we omit the algorithm for `NAF`, whose structure would be very similar to this one. The key difference is that `NAF` employs only one network and all the gradients are computed from the Q-learning target.

## SUPPLEMENTARY REFERENCES

[S1] P. Chou, D. Maturana, and S. Scherer. Improving stochastic policy gradients in continuous control with deep reinforcement learning using the beta distribution. In *International Conference on Machine Learning*, pp. 834–843, 2017.

[S2] X. Glorot and Y. Bengio. Understanding the difficulty of training deep feedforward neural networks. In *Proceedings of the 13th international conference on artificial intelligence and statistics*, pp. 249–256, 2010.

[S3] S. Gu, T. Lillicrap, Z. Ghahramani, R. E Turner, and S. Levine. Q-prop: Sample-efficient policy gradient with an off-policy critic. In *International Conference on Learning Representations (ICLR)*, 2017.

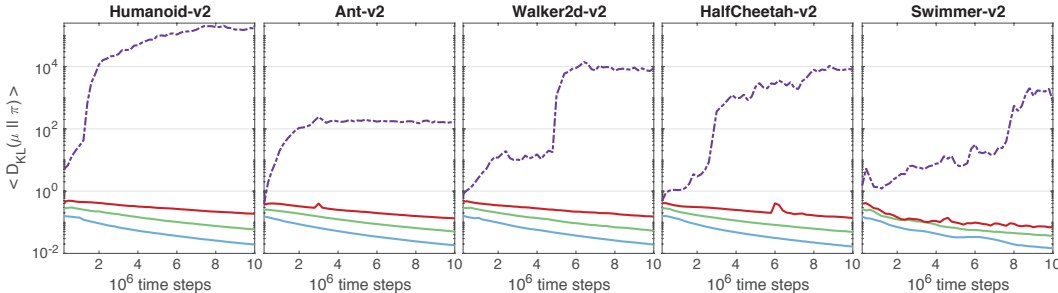

Figure 7: Average $D_{KL}(\mu_t\|\pi^{\mathtt{w}})$ for `ACER` (purple line), `RACER` baseline (green line, $C=4$, $N=10^{19}$), `RACER` with conservative `ReF-ER` constraints and large RM size (blue line, $C=2$, $N=10^{20}$), and `RACER` with relaxed `ReF-ER` constraints and smaller RM size (red line, $C=8$, $N=10^{18}$).

[S4]  D. Kingma and J. Ba. Adam: A method for stochastic optimization. In *International Conference on Learning Representations (ICLR)*, 2015.

[S5]  K. B. Petersen, M. S. Pedersen, et al. The matrix cookbook. *Technical University of Denmark*, 7(15):510, 2008.

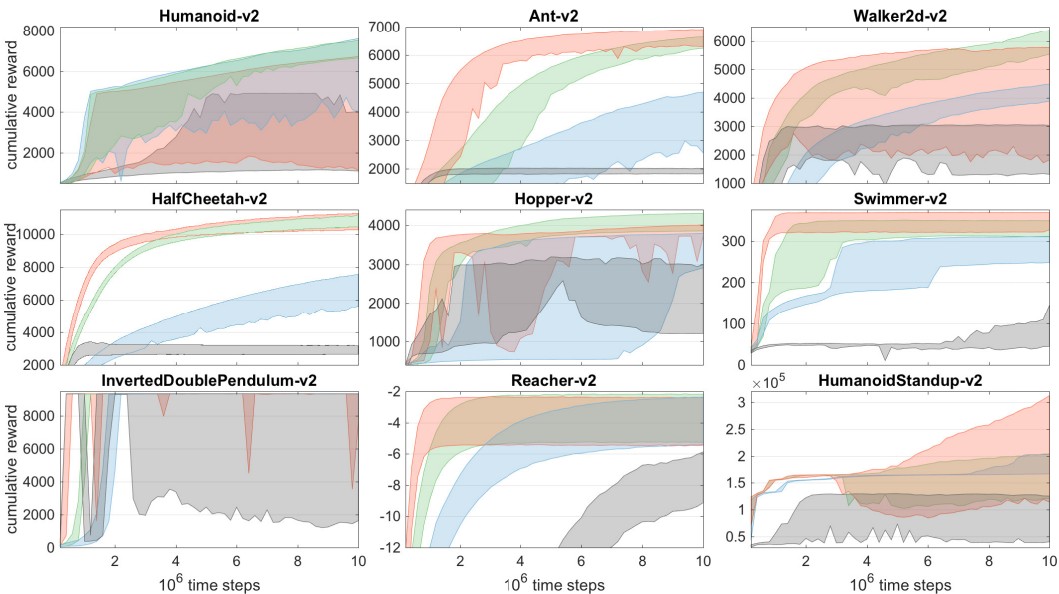

Figure 8: 20$^{th}$ and 80$^{th}$ percentiles of cumulative rewards for episodes of OpenAI Gym tasks ended within intervals of $2 \cdot 10^5$ time steps across 5 independent training runs. The figures includes results for `PPO` (black contours), `RACER` baseline (green contours, $C = 4$, $N = 10^{19}$), `RACER` with conservative `ReF-ER` constraints and large RM size (blue contours, $C = 2$, $N = 10^{20}$), and `RACER` with relaxed `ReF-ER` constraints and smaller RM size (red contours, $C = 8$, $N = 10^{18}$).

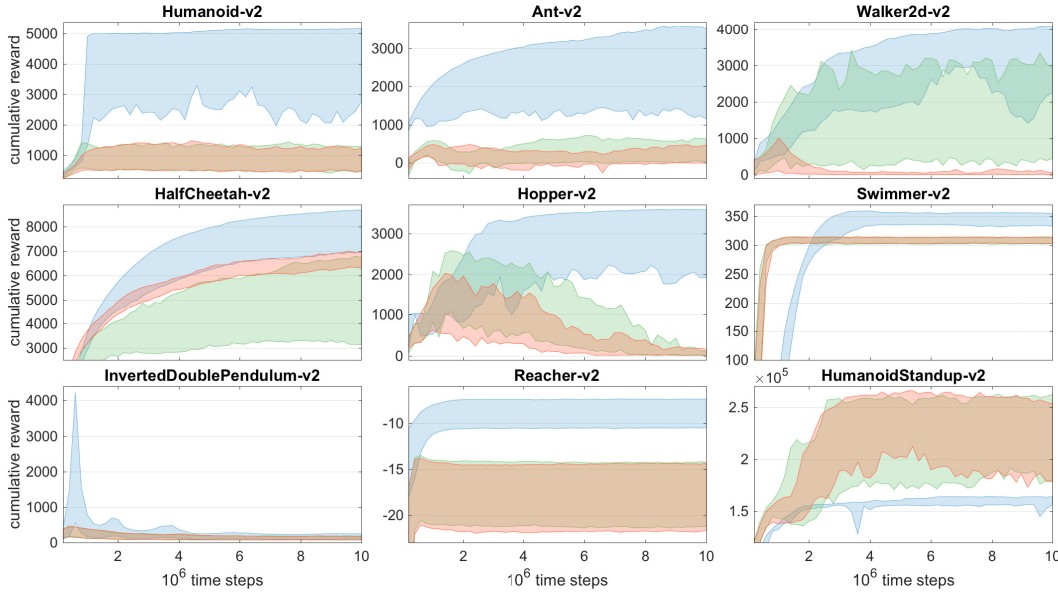

Figure 9: 20$^{th}$ and 80$^{th}$ percentiles of cumulative rewards for episodes of OpenAI Gym tasks ended within intervals of $2 \cdot 10^5$ time steps across 5 independent training runs. The figures includes results for `DDPG` (green contours), `DDPG` with `PER` (red contours), and `u-DDPG` with `ReF-ER` (blue contours, $C = 4$, and all other hyper-parameters set according to App. D).

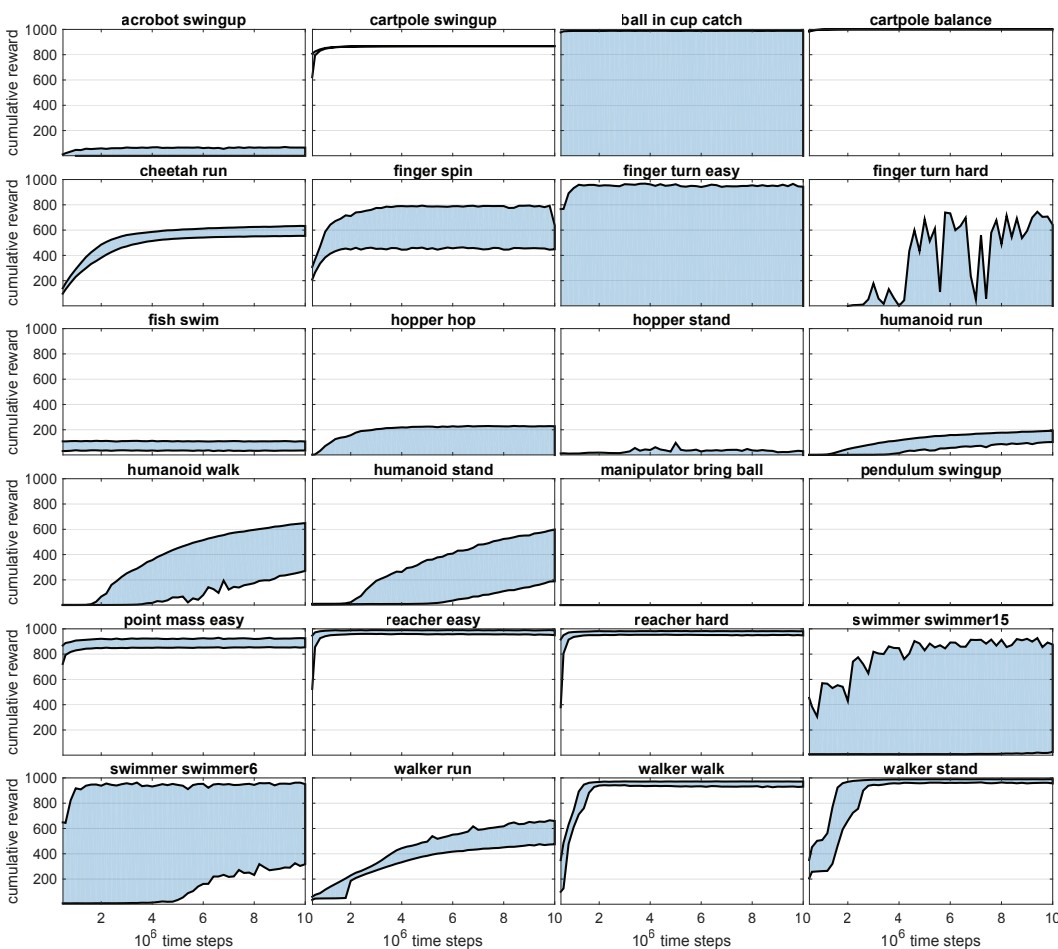

Figure 10: 20th and 80th percentiles of cumulative rewards for trajectories of DeepMind Control Suite tasks ended within intervals of $2 \cdot 10^5$ time steps across 5 independent training runs using the baseline RACER hyper-parameters (Table 1).

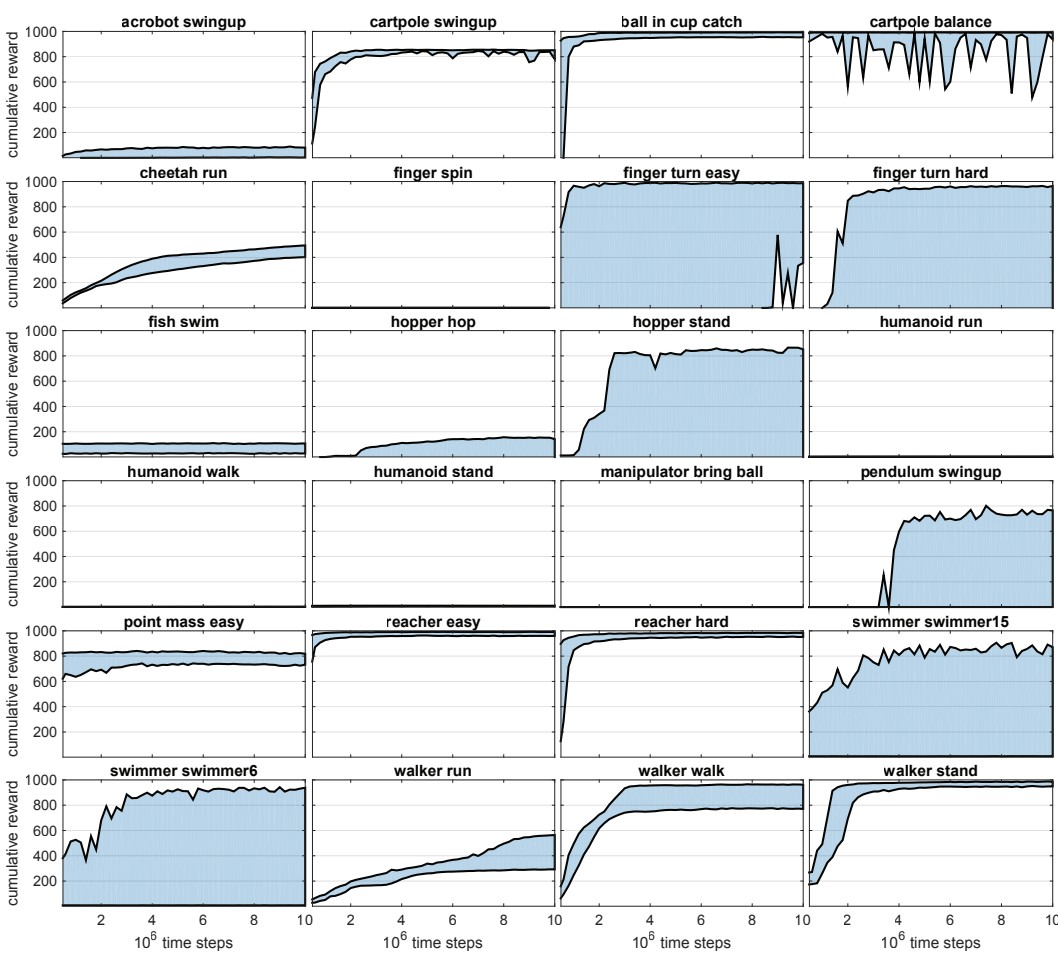

Figure 11: $20^{\text{th}}$ and $80^{\text{th}}$ percentiles of cumulative rewards for trajectories of DeepMind Control Suite tasks ended within intervals of $2 \cdot 10^5$ time steps across 5 independent training runs for `u-DDPG` with `ReF-ER`, $C = 4$, and all other hyper-parameters set according to App. D.

