# OpenReview forum: "Remember and Forget for Experience Replay"
_ICLR.cc/2019/Conference_

### Official Review · AnonReviewer2 · 2018-11-02

**Rating:** 6
**Confidence:** 3

**Review:**

This paper first proposed a variant of experience replay to achieve better data efficiency in off-policy RL. The RACER algorithm was then developed, by modifying the approximated advantage function in the NAF algorithm. The proposed methods were finally tested on the MuJoCo environment to show the competitive performance.

This paper is in general well written. The ideas look interesting, even though they are mostly small modification of the previous works. The experiments also show the promise of the proposed methods. One of my concerns is regarding the generality of ReF-ER. I am wondering if it can be also applied to the Atari domain to boost the performance there, similar to the prioritized experience replay paper. I understand that the requirement of GPUs is beyond the hardware configuration in this work, but that would be an important contribution to the community. My other questions and comments are as follows.
- Regarding the parametric form of f^w in Eq. (7), what are the definitions for L_+ and L_-? What are the benefits of introducing min and max there, compared with the form in Eq. (11), as used in NAF? Does it cause any problems during optimization?
- The y axis in Figure 3 is for KL (\pi || \mu), while the text below used KL(\mu || \pi) and the description regarding the change of C also seems to be inaccurate.
- In Figure 4, do you have any explanation why using PER leads to worse performance for NAF?
- For the implementation, did you use any parallelization to speed up the algorithm?

---

> ### Author Response · Authors · 2018-11-15
> **Author response to AnonReviewer2**
>
> We thank the Reviewer for the comments and careful reading of our work.
>
> > The ideas look interesting, even though they are mostly small modification of the previous works.
>
> We respectfully disagree with this assessment. We believe we have made the following novel contributions:
> 1) We presented a new way to quantify and account for the relevance of off-policy experiences. It is commonly believed that off-policy methods (e.g. Q-learning) can handle the dissimilarity between off-policy and on-policy outcomes. We provide ample evidence that training from highly similar-policy experiences is essential to the success of off-policy continuous-action deep RL.
> 2) We present a technique for Experience Replay that, without changes to its hyper-parameters, is shown to benefit continuous action deep RL across multiple specific methodologies (Q-learning, DPG, off-policy PG). The present method is modular and can be applied to many RL methods with parameterized policies.
> 3) We introduce an off-policy PG architecture that achieves state-of-the-art results in continuous-action benchmarks. This method is also far less computationally demanding than other off-policy PG methods (e.g. ACER or IMPALA).
>
> > The experiments also show the promise of the proposed methods. One of my concerns is regarding the generality of ReF-ER. I am wondering if it can be also applied to the Atari domain to boost the performance there, similar to the prioritized experience replay paper. I understand that the requirement of GPUs is beyond the hardware configuration in this work, but that would be an important contribution to the community.
>
> The generality of ReF-ER is indeed demonstrated only for continuous action RL. We focus on the continuous-action methods, as they are arguably brittle and dependent on hyper-parameters (Islam et al. 2017, Henderson et al. 2017). Applications to the Atari domain implies demonstrating the validity of ReF-ER to RL with discrete actions. While we very much value this extension, at present we see two difficulties: 1) The number of RL methods to consider and compare against would likely double, making for a much longer paper. 2) The computational resources required to perform the study would dramatically increase.
>
> We modified the Conclusions stating that extensions of ReF-ER to discrete action RL are a subject of ongoing work.
>
> > - Regarding the parametric form of f^w in Eq. (7), what are the definitions for L_+ and L_-? What are the benefits of introducing min and max there, compared with the form in Eq. (11), as used in NAF? Does it cause any problems during optimization?
>
> We compared several definitions of the action Advantage (Aw) (Eq. 7) in the Appendix. We observed that a symmetric parametrisation of the Aw led to worse results. Especially in tasks like HalfCheetah, where the agent has to balance control penalties and the optimality of a "bang-bang" controller, the return landscape is likely to be asymmetric around the mean of the policy.
>
> > - The y axis in Figure 3 is for KL (\pi || \mu), while the text below used KL(\mu || \pi) and the description regarding the change of C also seems to be inaccurate.
>
> We are grateful to the reviewer for finding these errors. We have amended them.
>
> > - In Figure 4, do you have any explanation why using PER leads to worse performance for NAF?
>
> An intuitive explanation is presented in section 6.2. The quadratic approximation of Aw performed by NAF is likely to be accurate in a small neighborhood around the mean m. For actions far from m, the approximated Aw may assume large negative values. PER samples more often time steps associated with greater TD errors. Hence, NAF tries to correct Aw for experiences that are more likely to be farther from the mean. This might not help the learner to fine-tune the policy by improving the local approximation of Aw.
>
> > - For the implementation, did you use any parallelization to speed up the algorithm?
>
> We used shared-memory parallelization to speed up the algorithm. Processing of the mini-batch is performed by multiple threads. The environment simulator runs on a separate process and a dedicated thread answers the queries of the simulation for actions. A fixed number of simulation time-steps is allowed per gradient step (usually one). We developed the methods with our in-house open-source C++ framework.

---

> > ### Comment · AnonReviewer2 · 2018-12-03
> > **Thanks for the response**
> >
> > I appreciate the authors' response, from which some of my concerns have been addressed. I still hold my view regarding the significance and theoretical novelty of this work, while I do believe that the empirical results could potentially benefit certain researchers in this field.

---

### Official Review · AnonReviewer1 · 2018-11-02
**REMEMBER AND FORGET FOR EXPERIENCE REPLAY REVIEW**

**Rating:** 6
**Confidence:** 3

**Review:**

This paper presents a method for forgetting and re-weighting experiences from a buffer during updates. It is well quantified experimentally and has some interesting tricks to improve performance in DDPG and other methods in continuous control which make use of a replay buffer. The authors also present another method “RACER” which makes use of this.

I would like to see this published at some point, particularly because of the interesting results on DDPG. However, while it is interesting and useful, I do I have concerns both on the novelty and experimental comparisons in the current version. For example, RACER seems similar to ACER, yet doesn’t compare to it, making it difficult to understand what is its benefit other than its use of REFER. Moreover, the authors state that without the REFER part (with PER instead), RACER doesn’t work well at all, making it difficult to assess the RACER algorithm on its own. I would suggest if the authors claim that the contribution is the REFER algorithm they assess REFER in ACER on its own to make the main contribution stronger.

Regarding novelty, I suggest that more of the paper can be spent situating the work in the broader scope of experience selection. There were several other methods that could have been compared against — for example (de Bruin et al., 2015) —which also presents a forgetting method similar to this one. While that work is cited, I don't believe it is sufficiently contrasted against this work.

Below I will examine various points/thoughts that came up.

+ well experimented, appreciated the use of confidence intervals in the appendix and extensive ablation. However, I’d like to point out that the confidence intervals for some tasks spanned anything from 0 to the max, which did not inspire confidence. However, this may be a problem with the task and not the method, so not a significant problem
+ clearly a lot of effort went into getting all these experiments and architecting the system which is well appreciated, great job there.
+ DDPG results are promising and may indicate the problem with DDPG is its off-policy-ness. Nice results there.
+ For the Re-Fer part, it was a bit unclear why it is 1/c_max < p_T <c_max rather than 0 < p_T < c_max? I suppose this is because you still want to update even if your current policy has not likelihood of that action? It would be nice to point to an explanation from that part of that text even if the intuition is in the appendix, otherwise it’s a bit unclear as to why this is chosen to be the acquisition function.
+ Along these lines it would be good to see more theoretical examination of on-policiness, rather than a binary threshold of the importance weight.
+ This paper seemed somewhat unfocused and packed with stuff, almost like two papers together which made things a bit difficult to follow as to what the main contribution is. I believe this detracted from both methods. For example, it was unclear what the benefit of using RACER was vs. say any other method which makes use of REFER. As the authors state, RACER without the ReFer part seems to not really work well at all, which makes me question this part of the contribution. It seems like a more interesting experiment would be to update importance weighted off-policy PG algorithms with the REFER part. This would hone the message which seems to be the main contribution of the paper.
+ I find it surprising that the authors compared PPO against RACER rather than using ACER which seems like the nearest analogue to this algorithm or IMPALA which seems to have a similar parallelized architecture.
+ More work could have been cited on experience selection selection, for example:

Isele, D., & Cosgun, A. (2018). Selective Experience Replay for Lifelong Learning. arXiv preprint arXiv:1802.10269.
Pan, Yangchen, Muhammad Zaheer, Adam White, Andrew Patterson, and Martha White. "Organizing Experience: A Deeper Look at Replay Mechanisms for Sample-based Planning in Continuous State Domains." arXiv preprint arXiv:1806.04624 (2018).

(I am aware that these are relatively new works, but after looking at the posting timestamps, I believe the original versions were posted several months at least prior to this publication.)

+ Along these lines I have concerns about the novelty since de Bruin 2015 even uses a similar off-policy metric for forgetting already. There are several differences here, but I’m not sure if they’re significantly novel for publication in its current state.

Typos/Grammar Issues Found:

“However, the information contained in consecutive steps is highly correlated, worsening the quality of the gradient estimate, and episodes can be composed of thousands of time step.” —> “However, the information contained in consecutive steps is highly correlated, worsening the quality of the gradient estimate, and episodes can be composed of thousands of time step(s).”

---

> ### Author Response · Authors · 2018-11-09
> **Author response to AnonReviewer1**
>
> We appreciate the Reviewer’s time, consideration and insightful comments. We extended the manuscript, added the suggested References and fixed typos. In the following we respond to all comments of the Reviewer. We hope that the paper in its current form is acceptable for publication.
>
> > RACER seems similar ... use of REFER.
>
> We now include the comparison with ACER. We find that both PPO and RACER outperform ACER.
>
> > Moreover, the authors ... contribution stronger.
>
> Our objective is to compare ER strategies and the 3 dominant methods of off-policy continuous actions RL: Q-learning (NAF), deterministic PG (DDPG), and stochastic PG (such as ACER or RACER). We agree that RACER is a secondary contribution but necessary to accurately analyze ReF-ER in the off-policy PG setting. We further clarify in the next paragraphs.
>
> NAF and DDPG are simple and widely used methods that train by sampling time steps. Conversely, ACER and IMPALA define expensive network architectures and train by sampling episodes. Their computational cost hinders reproducibility and exploration of hyper-parameters. We are not aware of any published work that compares against these methods for continuous-control.
>
> Moreover, we cannot analyze ReF-ER by coupling it with ACER. ACER introduces techniques that would interfere with ReF-ER. a) ACER bounds policy updates in a trust-region of a target network. b) ACER uses the "truncation and bias correction trick" to bound \rho. c) ACER modifies the value of \rho employed by Retrace (\rho_t becomes \rho_t^{1/d_A}), with unknown effects on the convergence guarantees of Retrace.
> These techniques were introduced to handle vanishing/exploding importance weights that are detrimental to off-policy PG.
>
> Removal of these techniques from ACER implies defining a new algorithm. This new algorithm differs from RACER by 2 features. RACER uses bootstrapped Retrace estimates (enabling training from times steps) and uses a closed-form Advantage (computing the advantage in ACER is expensive and approximate). In order to handle unbounded \rho, RACER requires ReF-ER (Sec 6.3). We show that ReF-ER and RACER achieve state-of-the-art results.
>
> In the revised manuscript we extend comparisons of RACER to ACER.
>
> > Regarding novelty, ... contrasted against this work.
>
> We extended the Related Work section to better position our paper.
>
> We note that de Bruin at al. (2015) focus on a simplified robotic task with 2D action space and short time horizon (30 actions to complete an episode). Due to the simplicity of the task, learning occurs over 30 trials and the RM holds 2 to 10 prior episodes. Both numbers are orders of magnitude smaller than what is required to solve MuJoCo tasks.
>
> The key result of de Bruin et al. related to ER is that keeping the first few acquired episodes in the RM throughout training stabilizes DDPG when the RM is very small. However, de Bruin et al. concede that their method is not applicable for longer training times because the networks will just overfit the few initial episodes. ReF-ER is not affected by this limitation.
>
> Finally, the method of de Bruin et al. increases the diversity of behaviors in the RM. This is opposite to the goal of ReF-ER and we respectfully disagree with the conclusions in de Bruin et al. We find that most off-policy RL methods benefit from computing updates with near-policy behaviors. If training from a diversity of behaviors were to be beneficial, ReF-ER would be outperformed by ER or PER.
>
> > For the Re-Fer part, it was a bit unclear why it is 1/c_max < p_T <c_max rather than 0 < p_T < c_max
>
> We focused on the importance weight \rho because it affects many off-policy estimators (e.g. the policy optimization objective and the Retrace algorithm). Our acquisition function prevents exploding (leading to instability) as well as vanishing (indicating irrelevance) \rho from affecting these estimates. Furthermore, for a given action, \rho deviates from unity if the KL distance between policy and RM behavior increases. We expand Section 3 to clarify these issues.
>
> > Along these lines it would be good to see more theoretical examination of on-policiness, rather than a binary threshold of the importance weight.
>
> We agree with the Reviewer. The main contribution of this work is that, controlling the degree of off-policiness can both stabilize training and improve the returns in continuous-action RL. We believe that the issue of off-policy learning is not properly attended in the RL community. We are not in a position to carry out this theoretical examination but hope that this work will inspire such efforts.

---

> > ### Comment · AnonReviewer1 · 2018-11-24
> > **Appreciate update/feedback**
> >
> > I appreciate the updated version of the paper and the feedback, it clears some of the issues up. While, I do still have some concerns regarding novelty and contributions as compared to more theoretically founded related work, the experimental results may be of interest to the community and as such I am updating my original rating.
> >
> > One thing I noticed in the new revision is that the supplementary references are interspersed by the figures currently, I would suggest that the authors try to wrangle Latex to make that a bit more readable.

---

### Official Review · AnonReviewer4 · 2018-11-07

**Rating:** 7
**Confidence:** 3

**Review:**

The authors introduce two new algorithms: remember and forget experience replay (ReF-ER), and an actor-critic architecture for continuous-action problems which is significantly more computationally efficient than previous approaches (RACER). ReF-ER manages the experience in the replay memory more directly and removes trajectories (episodes) that follow policies less related to the current parameterized policy (based on the importance weights). RACER's main contribution is provides a closed form approximation of the action values, enabling significant gains computationally. They provide several empirical studies in benchmark domains showing the competitiveness of their approach, and the provided more stability to various continuous control algorithms (NAF, PG, u-DDPG).

Overall, I think it is a nicely written paper with a lot of empirical evidence of the usefulness of ReF-ER. I am quite interested in this algorithm specifically, as the active management of experience in the replay memory is an important step towards the ER acting as a proxy to short term memory. To my knowledge this algorithm is novel, and performs admirably. I'm less clear of the main benefits of RACER over previous approaches, except for better computational complexity. This primarily comes from a lack of empirical comparison, and not much explanation as to why key competitors were excluded. The inclusion of RACER seems to muddy the message of the paper, and a much stronger and deeper look at ReF-ER would have made for a stronger submission.

I have several questions for clarity and more comments below, but overall I think the paper is quite useful for the community and contains interesting insight into active management of transitions in an experience replay buffer.

Pros:
------

Lots of empirical studies. And a lot of details to impart intuition of the new experience replay.

Interesting take on experience replay.

Convincing results in many simulation benchmark domains (even though the competitors are sparse).

Cons:
------

There is some ambiguity and maybe some confusion about the difference between control and off-policy learning. While I agree you are learning off-policy for control (due to the experience replay buffer containing old data), the terms off-policy and on-policy seem overused here. Statements such as "ER has become one of the mainstay techniques to improve the sample-efficiency of off-policy RL" aren't entirely correct as the experience replay buffer is primarily used in deep reinforcement learning to improve sample-efficiency, not off-policy reinforcement learning as a whole.

The RACER algorithm seems to muddy up the message of the paper quite a bit. I would have much preferred an in-depth look at ReF-ER here, rather than the introduction of two algorithms. And I think your paper would have been stronger for it. That being said, the RACER algorithm seems incomplete. While it is an improvement over prior approachers (ACER) computationally, the need to use ReF-ER is concerning. I'm also a bit confused why ACER isn't used as a competitor against RACER? Even if you aren't outperforming the other approach on all benchmarks, the improved computational complexity is still a worthwhile improvement.

No confidence bounds in the results, although these are somewhat shown in the appendix (without the competitors shown!!). I'm curious at the significance of the different parameter settings.

Questions:
----------

I'm curious as to how this is related to something like rejection sampling? Or other importance sampling approaches more directly? How does your method compare with using retrace or some other off-policy algorithms? I'm unclear on the reasons why these types of comparisons aren't made empirically, could you clarify more directly?

Does your algorithm help with variance issues of other off-policy algorithms? Such as just using importance weights instead of retrace? How would it effect tree backup or just the usual importance sampling? It seems likely that this would help here, as you are limiting the amount of data with high importance weights, although this might also add bias.

Have you removed the target network in your experiments? This detail is not obvious in the paper currently and when you introduce ReF-ER you seem to be leading to this, but never say explicitly.

You claim that ReF-ER "reduces the sensitivity on the network architecture and training hyper-parameters." I'm unclear how you show this in the results with the current paper. You do some hyperparameter studies in the appendix, but don't compare against other algorithms here. Could you share a bit further how you are measuring the sensitivities of your algorithm against the competitors?

Do you need to anneal the cmax? What are the effects if this is set to some constant?

Could you expand on the results of HumanoidStandup-v2? Why do you believe your approach does significantly worse than the baselines here?

For DDPG, what happens if you change the bounds instead of removing them entirely? Also how does your method compare on a domain without unbounded actions?

It is unclear why RACER does not work with ER/PER. Do you have any intuition here? Could this be fixed through means other than ReF-ER?


Other minor comments (not taken into consideration for the review):
-------

Pseudo code: It is a bit unclear what algorithm 1 is supposed to be, I'm assuming ReF-ER?


Begin revision comments:
-----

Given the revisions to this paper, I am more confident that it will be of interest to the community. The major contributions here I see is the removal of target networks given their approach. Given this I still have concerns on clarity and still am unhappy with the lack of confidence intervals in the main experimental section. I've increased my score to 7 to reflect my increase in confidence.

---

> ### Author Response · Authors · 2018-11-15
> **Author response to AnonReviewer4**
>
> We thank the Reviewer for the encouraging comments and useful feedback. We revised the paper to include more evidence of the benefits of our approach compared to key competitors (please see Sec. 6.3 and Fig. 7 to 9).
>
> > The inclusion of RACER ... stronger submission.
> We have compared ReF-ER to the most widely used techniques for ER. We find that ReF-ER, without changes to its hyper-parameters, benefits continuous action deep RL across multiple methodologies.
>
> We could have coupled ReF-ER to ACER rather than introducing RACER. However, the key techniques of ACER have the purpose of dealing with unbounded importance weights. These techniques interfere with ReF-ER as they accomplish a similar goal: control policy changes.
> Removing these features from ACER would imply a new algorithm that would be similar to RACER albeit with added computational cost (due to the sampling-based advantage estimation and due to training based on episodes).
>
> > While it is an improvement ... against RACER?
> RACER without ReF-ER does not have measures to deal with unbounded \rho, enabling a clearer analysis of ReF-ER’s hyper-parameters (HP). In the absence of ReF-ER, unbounded \rho cause off-policy PG to be unstable.
>
> We now include a comparison showing that RACER (and PPO) outperform ACER.
> The results for ACER are obtained by keeping most HP as in the original (App. D). We removed the convolutions as here learning is not from pixels. We tuned the learning rate, the target-network update, and the layers' size. It is known that HP can affect performance (Henderson et al. 2017) and we do not exclude that ACER could have performed better with more extensive tuning. Fig. 7 shows that, despite our tuning, ACER's techniques do not prevent \pi from diverging from past behaviors.
>
> We have extended Sec. 4 to further motivate introducing RACER.
>
> > There is some ambiguity ... as a whole.
> We corrected the manuscript to clearly state deep RL wherever appropriate.
>
> > I'm curious as to how ... approaches more directly?
> Algorithms that couple the off-policy PG and ER (e.g ACER, RACER) perform importance sampling. The policy is evaluated from the distribution of experiences in the RM, and we account for actions being off-policy with the importance weights \rho.
> A more direct application of rejection sampling would imply computing a constant C such that:
> P( sample | ER ) * C >= P( sample | on policy )
> We find this comment very intriguing to pursue further and we welcome references.
>
> > How does your ... off-policy algorithms?
> ReF-ER is “orthogonal” to Retrace. In fact, RACER uses Retrace to provide a target for value learning. We would welcome further questions if we misunderstood the Reviewer’s comment.
>
> > Does your algorithm ... add bias.
> We do not have results of value estimators without the truncation introduced by Retrace. We suspect that the product of many \rho may be unstable even with strict ReF-ER constrains (e.g. C=2). We intend to investigate this issue with further experiments.
>
> > Have you removed … say explicitly.
> We do not use target-networks for RACER and they are not required by ReF-ER.
>
> > You claim … competitors?
> We amended the relevant sentence to clarify that it describes an objective of the design of ReF-ER.
>
> > Do you need to ... constant?
> Annealing cmax leads to better performance. It allows ReF-ER to bridge the gap between off-policy RL and on-policy methods like PPO. At the end of training, the experiences in the RM are constrained to be almost on-policy.
>
> > Could you expand ... baselines here?
> Our understanding is that the HumanoidStandup task requires the agent to flail around to acquire vertical momentum. Fig.7 suggests that this task is easier to learn with a small RM and relaxed ReF-ER constrains, allowing the agent to explore with faster policy changes.
>
> > For DDPG, ... unbounded actions?
> We tested on the DeepMind suite which bounds the control space to [-1 1]. There are several ways to deal with the bounded domain. We employed the simplest: advance the algorithm as if actions were unbounded, then map to the control space with a tanh.
> This approach is naive, but deviates the least from the methods we employed in the other benchmarks.
>
> To understand why this approach performs poorly we propose an example. A policy with mean in 0.999 and one with mean in 1 may produce similar outcomes, but their KL divergence in unbounded space may be very large. An alternative is to employ the Beta distribution (Chou et al., 2017) which is also compatible with ReF-ER.
>
> > It is unclear why RACER ... other than ReF-ER?
> An alternative would be to employ the techniques of ACER (eg. "truncation and bias correction trick", target networks and modified Retrace). We thought of comparing RACER with ReF-ER to RACER with ER and these techniques. However, each one introduces one or more HP, requiring optimization and increasing the complexity of the paper.
>
> > Pseudo code: It is a bit ... ReF-ER?
> We amended the manuscript to clarify the appendix.

---

> > ### Comment · AnonReviewer4 · 2018-11-27
> > **Good revisions, answered concerns**
> >
> > I've updated my review above, and increased my score given these additions.

---

### Meta-Review · Area_Chair1 · 2018-12-13
**Some interesting ideas with a bit more work needed to understand the importance of each component**

**Confidence:** 3
**Recommendation:** Reject

**Metareview:**

This paper introduces a novel idea, and demonstrates its utility in several simulated domains. The key parts of the algorithm are (a) to prefer keeping and using samples in the ER buffer where the corresponding rho_t, using the current policy pi_t, are not too big or small and (b) preventing the policy from changing too quickly, so that samples in the ER buffer are more on-policy.

They key weakness is not better investigating the idea of making the ER buffer more on-policy, and the effect of doing so. The experiments compare to other algorithms, but do not sufficiently investigate the use of both Point 1 and Point 3. Further, the appendix contains an investigation into parameter sensitivity and gives some confidence intervals. However, the presentation of this is difficult to follow, and so it is difficult to gauge the sensitivity of Ref-ER. With a more thorough experimental section, better demonstrating the results (not necessarily running more things), the paper would be much stronger.

For more context, the authors rightly mention "It is commonly believed that off-policy methods (e.g. Q-learning) can handle the dissimilarity between off-policy and on-policy outcomes. We provide ample evidence that training from highly similar-policy experiences is essential to the success of off-policy continuous-action deep RL." Q-learning can significantly suffer from changing the state-sampling distribution. However, adjusting sampling in the ER buffer using rho_t does not change the state-sampling distribution, and so that mismatch remains a problem. Changing the policy more slowly (Point 3) could help with this more. In general, however, these play two different roles that need to be better understood. The introduction more strongly focuses on classifying samples as more on or off-policy, to solve this problem, rather than the strategy used in Point 3. So, from the current pitch, its not clear which component is solving the issues claimed with off-policy updates.

Overall, this paper has some interesting results and is well-written. With more clarity on the roles of the two components of Ref-ER and what they mean for making the ER buffer more on-policy, in terms of both action selection and state distribution, this paper would be a very useful contribution to stable control.